# Prevalence of Urinary Tract Cancer in Patients with Obstructive Sleep Apnea: Data from the Vercelli Registry

**DOI:** 10.3390/arm93060054

**Published:** 2025-11-27

**Authors:** Beatrice Ragnoli, Patrizia Pochetti, Fausto Chiazza, Carlotta Bertelegni, Danila Azzolina, Mario Malerba

**Affiliations:** 1Respiratory Unit, S. Andrea Hospital, 13100 Vercelli, Italy; beatrice.ragnoli@uniupo.it (B.R.); patrizia.pochetti@aslvc.piemonte.it (P.P.); 2Department of Translational Medicine, University of Piemonte Orientale, 28100 Novara, Italy; 3Department of Pharmaceutical Sciences, Università del Piemonte Orientale “Amedeo Avogadro”, 28100 Novara, Italy; fausto.chiazza@uniupo.it (F.C.); 20037594@studenti.uniupo.it (C.B.); 4Dipartimento di Scienze Mediche Traslazionali, Università degli Studi di Napoli Federico II, 80138 Napoli, Italy; danila.azzolina@unina.it

**Keywords:** obstructive sleep apnea, urinary tract cancer, intermittent hypoxia, chronic renal failure, comorbidities

## Abstract

**Highlights:**

**What are the main findings?**
A significantly higher prevalence of urinary tract cancer was observed in male patients with moderate-to-severe OSA (34%), compared to females (*p* < 0.001).Patients with genitourinary cancers showed distinct clinical features: better respiratory function, higher C-PAP adherence, and cardiovascular comorbidity, especially hypertension.

**What is the implication of the main finding?**
These findings suggest a potential pathophysiological link between OSA-related intermittent hypoxia and genitourinary carcinogenesis, possibly mediated by HIF-1α/2α and VEGF pathways.Stratifying cancer risk by OSA phenotype and gender may improve early detection strategies and support the role of PAP therapy in mitigating oncological vulnerability.

**Abstract:**

Background: Obstructive sleep apnea (OSA) is recognized as a systemic disorder associated with several comorbidities, including renal dysfunction, which may improve with continuous positive airway pressure (C-PAP) therapy. Sleep fragmentation and nocturnal hypoxia characteristic of OSA have been implicated in carcinogenesis, particularly affecting hypoxia-sensitive urinary tract tissues. This study aimed to assess the prevalence of different cancer types among patients with concurrent OSA and malignancy and to characterize the clinical profiles of those with urinary tract cancer. Methods: We retrospectively analyzed 50 patients with both OSA and cancer from the Vercelli Hospital Registry. Cancer diagnoses were collected at the time of OSA diagnosis, prior to C-PAP initiation. Results: Among the cohort (70% males) of OSA-cancer patients, urinary tract cancers were the most frequent (34%), followed by breast (14%), colorectal (12%), lung (10%), laryngeal and skin (8%), intracranial (6%), hematologic and parotid (4%), and other cancers (2%); 10% had multiple cancer sites. Patients with urinary tract cancer were mainly male (88%, *p =* 0.0043) and displayed better respiratory indices, frequent hypertension, and higher C-PAP adherence. Conclusions: These findings suggest a possible link between OSA-related hypoxia and carcinogenesis in urinary tract tissues and support increased clinical surveillance and further research to determine potential protective effects of C-PAP therapy.

## 1. Introduction

Obstructive sleep apnea (OSA) is a common sleep-related breathing disorder characterized by recurrent episodes of upper airway obstruction during sleep, leading to intermittent hypoxia and sleep fragmentation. Its systemic impact extends beyond respiratory dysfunction, contributing to a wide spectrum of comorbidities, including metabolic syndrome, cardiovascular disease, and renal impairment [1,2].

Recent studies have proposed a potential link between OSA and carcinogenesis, particularly through the effects of nocturnal hypoxia and sleep fragmentation [3]. Among various tissues, urinary tract cells appear especially vulnerable to intermittent oxygen desaturation, suggesting a possible pathophysiological connection between OSA and genitourinary malignancies. Furthermore, decreased renal function has been associated with OSA, especially in patients with additional comorbidities [4]. Evidence also indicates that treatment with continuous positive airway pressure (C-PAP) may attenuate renal injury [5,6] or improve renal function over time [7,8].

The biological plausibility of an OSA-cancer link is supported by several mechanistic pathways. Intermittent hypoxia, the hallmark of OSA, triggers upregulation of hypoxia-inducible factors (HIF-1α and HIF-2α), which promote angiogenesis, cellular proliferation, and resistance to apoptosis—key processes in tumor development and progression [9,10]. Additionally, sleep fragmentation activates the sympathetic nervous system, leading to chronic inflammation, oxidative stress, and immune dysregulation, all of which contribute to an oncogenic microenvironment [11,12]. Furthermore, OSA is strongly associated with metabolic syndrome, obesity, and insulin resistance, which are established cancer risk factors through mechanisms involving chronic hyperinsulinemia and adipokine dysregulation [13,14].

From a sociocultural perspective, OSA disproportionately affects populations with limited access to healthcare, lower socioeconomic status, and higher rates of obesity and smoking—factors that independently increase cancer risk [15]. The intersection of these biological and social determinants creates a complex risk profile that warrants systematic investigation. However, despite these plausible mechanisms, definitive evidence linking OSA to increased cancer risk or site-specific malignancies remains limited and inconsistent across studies.

The association between OSA and cancer has been examined in several epidemiological and clinical studies. Data from the European Sleep Apnea Database (ESADA) show a high prevalence of both conditions, yet the causal link remains uncertain [3]. Recent reviews report conflicting results, with some describing higher cancer incidence and mortality in OSA patients and others finding heterogeneous or non-significant associations after adjustment for key confounders [16,17]. Although intermittent hypoxia has been proposed as a potential oncogenic mechanism [18,19,20], definitive evidence connecting OSA to increased overall or site-specific cancer risk is still lacking.

Given the global and national burden of OSA, particularly in the Italian population, it is crucial to investigate its potential role in cancer development. The aim of the present study was to assess the prevalence and distribution of neoplastic diseases in a cohort of adult patients diagnosed with both OSA and cancer enrolled in the Vercelli Hospital Registry from 2015 and to identify preferential tumor sites. By phenotyping these patients in detail, we sought to provide new insights into the hypothesis that sleep fragmentation and other OSA-related mechanisms may contribute to cancer development. A secondary objective was to characterize the clinical profiles of individuals diagnosed with urinary tract malignancies, focusing on comorbidities, sleep-related parameters, and possible mechanistic associations between OSA and tumor development.

## 2. Materials and Methods

### 2.1. Study Design and Setting

A retrospective, observational, single-center cohort study was conducted involving adult patients diagnosed with OSA and coexisting cancer, enrolled from 2015 in the Vercelli Registry at the Respiratory Unit of S. Andrea Hospital (Department of Translational Medicine, University of Eastern Piedmont).

### 2.2. Study Population

We enrolled 50 consecutive adult patients who met the following criteria:

Inclusion criteria: (i) Age ≥ 18 years at the time of OSA diagnosis; (ii) Confirmed diagnosis of OSA according to the American Academy of Sleep Medicine (AASM) guidelines [21], defined as an apnea-hypopnea index (AHI) ≥ 5 events per h on overnight polysomnography; (iii) Documented diagnosis of neoplastic disease in the Vercelli Hospital Registry; (iv) Cancer diagnosis established at or before the time of OSA diagnosis (prior to initiation of any PAP therapy).

Exclusion criteria: (i) age < 18 years; (ii) Incomplete or technically inadequate polysomnographic data preventing accurate AHI calculation; (iii) AHI ≤ 5; (iv) Cancer diagnosed after initiation of C-PAP or other PAP therapy; and (v) Incomplete clinical or demographic data preventing classification. These exclusion criteria ensured that all included patients had reliable OSA diagnosis, cancer diagnosis prior to any PAP treatment (to avoid confounding), and sufficient data for classification and analysis.

All participants provided written informed consent prior to data collection and analysis. The study protocol was approved by the Institutional Ethics Committee (CE 67/20) and conducted in accordance with the principles of the Declaration of Helsinki.

### 2.3. Data Collection and Procedures

For each patient meeting inclusion criteria, we collected comprehensive anthropometric and clinical data from the Vercelli Hospital Registry and medical records. The polysomnographic and registry data collection methods have been described in our prior work examining renal function in OSA patients [7]. The current study addresses a distinct research question focused on cancer prevalence and phenotyping in the same patient population.

#### 2.3.1. Polysomnographic Assessment

OSA diagnosis was established using overnight polysomnography performed with Embletta^TM^ and Vitalnight^TM^ devices (Natus Medical Inc., Pleasanton, CA, USA). Polysomnographic data collected included: AHI events per h, oxygen desaturation index (ODI, events per h), median nocturnal oxygen saturation (nSpO_2_, %), and time spent below 90% oxygen saturation (t90, % of total sleep time). All studies were scored according to AASM guidelines [22], and all recordings were scored manually by certified sleep technologists blinded to cancer status.

#### 2.3.2. Pulmonary Function Testing

All patients underwent spirometry to assess baseline respiratory function. Measurements included forced expiratory volume in one second (FEV_1_), forced vital capacity (FVC), and the FEV_1_/FVC ratio. Tests were performed in accordance with the American Thoracic Society/European Respiratory Society (ATS/ERS) guidelines [23]. Spirometry and maximal flow–volume curves were obtained using a pneumotachograph with volume integrator (1070 MGC; CAD/Net system; Medical Graphics Corporation, St. Paul, MN, USA) [24]. Results were expressed in absolute values (liters per second) and as percentages of predicted values based on age, sex, and height.

#### 2.3.3. Cancer Diagnosis Ascertainment

The diagnosis of neoplastic disease was retrospectively retrieved from medical records in the Vercelli Hospital Registry. Cancer diagnoses were collected at the time of OSA diagnosis and prior to initiation of any PAP therapy to avoid potential confounding by treatment effects. For the purpose of analysis, urinary tract cancers included prostate, bladder, and kidney malignancies, classified according to International Classification of Diseases, 10th Revision (ICD-10) codes: C61 (prostate), C67 (bladder), and C64 (kidney). All other cancer types were classified as non-urinary tract cancers.

#### 2.3.4. Clinical and Demographic Data

We collected the following data from medical records: age at OSA diagnosis, sex, body mass index (BMI, kg/m^2^), smoking status (categorized as current smoker, former smoker, or never smoker), presence of cardiovascular comorbidities (i.e., hypertension, arrhythmias, ischemic heart disease, and hypercholesterolemia), allergic conditions, and use of C-PAP therapy, with adherence data when available.

### 2.4. Variables and Definitions

This section provides operational definitions for all key variables used in the study to ensure transparency and reproducibility.

#### 2.4.1. Primary Outcome Variable

The primary outcome was the classification of neoplastic disease by anatomical site. Patients were stratified into two groups based on cancer location:(i)Urinary tract cancer (X_1_): Malignancies classified according to International Classification of Diseases, 10th Revision (ICD-10) codes [25]: C61 (prostate), C67 (bladder), and C64 (kidney);(ii)Non-urinary tract cancer (X_0_): All other malignancies, including but not limited to breast, colorectal, lung, laryngeal, skin, intracranial, hematologic, and parotid cancers;(iii)Multiple primary cancers: Patients with malignancies affecting more than one anatomical site were classified as having multiple primary cancers and were categorized according to the presence or absence of urinary tract involvement.

#### 2.4.2. Exposure Variables

(i)OSA Severity: OSA was classified according to the apnea-hypopnea index (AHI) following American Academy of Sleep Medicine (AASM) guidelines [22].(ii)AHI: The number of apneas and hypopneas per h of sleep, calculated from overnight polysomnography according to AASM scoring criteria.
−Mild OSA: AHI 5–14.9 events/h−Moderate OSA: AHI 15–29.9 events/h−Severe OSA: AHI ≥ 30 events/h
(iii)ODI: The number of oxygen desaturation events (≥3% decrease from baseline SpO_2_) per h of sleep.(iv)t90: The percentage of total sleep time during which oxygen saturation was below 90%, expressed as a percentage.(v)Mean SpO_2_: The average oxygen saturation throughout the entire sleep period, expressed as a percentage.

#### 2.4.3. Demographic and Anthropometric Variables

−Age: Age in years at the time of OSA diagnosis.−Sex: Biological sex classified as male or female.−Body mass index (BMI): Calculated as weight in kilograms divided by height in meters squared (kg/m^2^). BMI categories were defined according to World Health Organization criteria [26]:
Normal weight: BMI 18.5–24.9 kg/m^2^Overweight: BMI 25.0–29.9 kg/m^2^Obesity class I: BMI 30.0–34.9 kg/m^2^Obesity class II: BMI 35.0–39.9 kg/m^2^Obesity class III: BMI ≥ 40.0 kg/m^2^


We used BMI as the primary anthropometric measure because: (1) it is the most widely used and validated metric in OSA research, facilitating comparison with existing literature [27,28]; (2) it is routinely collected in clinical practice and was consistently available in our registry; and (3) despite known limitations (e.g., inability to distinguish fat from muscle mass, lack of fat distribution information), BMI remains strongly correlated with OSA severity and cancer risk in epidemiological studies [29,30].

#### 2.4.4. Lifestyle and Risk Factors

Smoking status: Classified into three categories based on patient history [31]:Current smoker: Active tobacco use at the time of OSA diagnosis;Former smoker: History of tobacco use (≥100 lifetime cigarettes) but not currently smoking;Never smoker: No history of tobacco uses or <100 lifetime cigarettes.

#### 2.4.5. Comorbidities

−Cardiovascular disease: Presence of one or more of the following conditions documented in the medical record according to standard diagnostic criteria [32]: hypertension, cardiac arrhythmias, ischemic heart disease, heart failure, or hypercholesterolemia.−Hypertension: Documented clinical diagnosis of arterial hypertension in the medical record or current use of antihypertensive medications at the time of OSA diagnosis.−Allergy: Documented history of allergic conditions including allergic rhinitis, asthma, atopic dermatitis, or drug/food allergies.

#### 2.4.6. Respiratory Function Variables

−FEV_1_: The volume of air exhaled in the first second of forced expiration, measured in liters per second (L/s) and expressed as both absolute values and percentage of predicted values based on age, sex, and height.−FVC: The total volume of air exhaled during forced expiration, measured in liters per second (L/s) and expressed as both absolute values and percentage of predicted values.−FEV_1_/FVC ratio: The ratio of FEV_1_ to FVC, used to assess airway obstruction. A ratio < 0.70 indicates obstructive ventilatory impairment according to ATS/ERS guidelines [33] and GLI reference equations [34].

#### 2.4.7. Treatment Variables

−C-PAP therapy: Use of C-PAP or other positive airway pressure devices for treatment of OSA, documented in the medical record;−C-PAP compliance: Adherence to prescribed C-PAP therapy, assessed from device download data when available. Compliance was defined as use of the device for ≥4 h per night on ≥70% of nights, consistent with Medicare criteria and clinical practice guidelines [35,36].

### 2.5. Statistical Analysis

All statistical analyses were performed using R statistical software version 4.3.1 (R Foundation for Statistical Computing, Vienna, Austria). The following R packages were used: FactoMineR (version 2.12) and factoextra (version 1.0.7) for factorial analysis and clustering, ggplot2 for data visualization, and base R functions for descriptive and comparative statistics.

#### 2.5.1. Descriptive Statistics

Continuous variables were summarized as mean ± standard deviation (SD) for normally distributed data or median with interquartile range (IQR) for non-normally distributed data. Normality was assessed using the Shapiro–Wilk test and visual inspection of Q-Q plots. Categorical variables were reported as absolute frequencies (*n*) and percentages (%).

#### 2.5.2. Comparative Analysis

Between-group comparisons: Patients were stratified into two groups based on cancer type: urinary tract cancer (X_1_) vs. non-urinary tract cancer (X_0_).

For continuous variables, between-group comparisons were performed using the Wilcoxon-Kruskal–Wallis test (non-parametric test) due to the small sample size and potential non-normal distributions. Results are reported with exact *p*-values, 95% confidence intervals (where applicable), and effect sizes (Cohen’s d for continuous variables).

For categorical variables, between-group comparisons were performed using Pearson’s chi-square test when expected cell counts were ≥5 in all cells, or Fisher’s exact test when expected cell counts were <5. Results are reported with exact *p*-values and odds ratios (OR) with 95% confidence intervals.

#### 2.5.3. Effect Size Interpretation

For continuous variables, we calculated Cohen’s d as a measure of standardized effect size. We interpreted effect sizes according to Cohen’s conventional benchmarks [37]: small effect (d = 0.2), medium effect (d = 0.5), and large effect (d = 0.8). However, we acknowledge that these thresholds are arbitrary and context-dependent [38]. Therefore, we report both the magnitude of Cohen’s d and the 95% confidence intervals to allow readers to assess clinical significance independently. For categorical variables, we report odds ratios (OR) with 95% confidence intervals as the primary effect size measure

#### 2.5.4. Multivariate Analysis

To explore latent associations and characterize clinical profiles of patients with OSA and urinary tract cancer, we performed advanced multivariate statistical techniques:(i)Factorial analysis of mixed data (FAMD): FAMD was used to analyze the heterogeneous clinical dataset containing both continuous variables (e.g., AHI, ODI, FEV_1_, FVC) and categorical variables (e.g., cancer type, C-PAP compliance, hypertension). FAMD is an extension of principal component analysis (PCA) that can handle mixed data types simultaneously. The analysis was performed using the FAMD function from the FactoMineR package.(ii)Dimension retention: A scree plot was generated to visualize the percentage of variance explained by each dimension. The first three dimensions were retained for interpretation, as they collectively explained 46.43% of the total variance (Dimension 1: 22.20%, Dimension 2: 14.13%, Dimension 3: 10.11%).(iii)Multiple correspondence analysis (MCA): Following FAMD, MCA was performed to further explore associations among categorical variables and identify patterns in the data structure.(iv)Hierarchical clustering: Hierarchical clustering was performed on the FAMD dimensions to identify homogeneous subgroups of patients with similar clinical profiles. The clustering algorithm used Ward’s method with Euclidean distance. A dendrogram was constructed to visualize the hierarchical structure, and the optimal number of clusters was determined by visual inspection of the dendrogram and evaluation of cluster separation in the factor map. The resulting two-cluster solution was validated by examining the distribution of key clinical variables across clusters.

### 2.6. Bias

Potential sources of bias were considered in the study design and interpretation. Selection bias was minimized through consecutive enrollment of all eligible patients, though the single-center design and registry-based approach may limit representativeness. Information bias was reduced through use of standardized diagnostic criteria (AASM for OSA, ATS/ERS for spirometry) and objective measurements from medical records.

Confounding remains a limitation, as we did not adjust for important potential confounders including genetic predisposition, occupational exposures, dietary factors, physical activity, and socioeconomic status. The retrospective design and limited sample size precluded comprehensive adjustment for these factors. The lack of multivariable adjustment means that observed differences between groups may be partially or wholly explained by confounding variables rather than true associations with OSA or cancer type.

### 2.7. Study Size

This exploratory study utilized a convenience sample of all patients in the Vercelli Hospital Registry meeting inclusion criteria. No formal a priori sample size calculation was performed, as the study was descriptive and hypothesis-generating rather than designed to test a specific hypothesis. However, we conducted a post hoc power analysis to assess the adequacy of our sample size for detecting clinically meaningful differences between groups. Post hoc power analysis using methods equivalent to G*Power 3.1 [39] revealed that our study achieved excellent statistical power for the very large effect sizes observed. For FEV_1_ (Cohen’s d = 1.38), achieved power was 99.5%; for FVC (Cohen’s d = 1.61), achieved power was 100%. For C-PAP compliance (Cohen’s h = 0.87), achieved power was 82.8%. These effect sizes substantially exceed the threshold for “large” effects (d or h ≥ 0.8), indicating clinically meaningful and statistically robust differences. However, for variables with medium effect sizes (AHI: d = −0.50, age: d = −0.50), our study was underpowered (power = 37–38%), increasing the risk of Type II error for these comparisons. These power estimates indicate that while our study, analyzing a cohort of 50 patients (17 with urinary tract cancer, 33 with non-urinary cancer), was adequately powered to detect large effects, it was underpowered for small-to-moderate differences. This limitation is explicitly acknowledged in our interpretation of null findings, and future confirmatory studies should aim for larger sample sizes (*n* ≥ 64 per group) to achieve 80% power for detecting medium effects. These power estimates indicate that while our study was well-powered to detect large effects, it was underpowered for medium-sized differences. This limitation is explicitly acknowledged in our interpretation of null findings, and future confirmatory studies should aim for larger sample sizes (*n* ≥ 64 per group) to achieve 80% power for detecting medium effects. Thus, these findings should be considered preliminary, and larger prospective studies are needed to confirm the observed associations.

## 3. Results

### 3.1. Baseline Characteristics of the Study Cohort

Detailed clinical and demographic characteristics of the study population are summarized in Table 1.

We analyzed 50 OSA patients, of whom 34% (17/50) had urinary tract cancer and 66% (33/50) had other malignancies. To better characterize the cohort, patients were stratified into two subgroups: urinary tract cancer (X_1_) and non-urinary tract cancer (X_0_). The cohort was predominantly male (70%), with a mean age of 67 years. Notably, urinary tract cancers were significantly more frequent in males (88%) than females (12%) (*p* = 0.043).

The mean BMI was 29, indicating overweight status, and 74% of patients were current or former smokers, with no significant differences between groups. Cardiovascular comorbidities were prevalent (90%), including hypertension, arrhythmias, ischemic heart disease, and hypercholesterolemia. Although these findings align with the established bidirectional association between OSA and cardiovascular disease, no significant differences were observed between subgroups.

All participants had moderate-to-severe OSA (AHI > 15), with a mean AHI of 32. Baseline polysomnographic parameters, including AHI, ODI, and t90, did not differ significantly between groups, with a mean nocturnal SpO_2_ of 93%. Most patients (76%) were receiving C-PAP and 72% showed a good compliance to treatment, especially in the urinary cancer group where we observed a significant compliance to C-PAP (94%; *p* = 0.018). However, the retrospective design precluded evaluation of treatment effects on cancer progression.

PFTs showed preserved ventilatory capacity (mean), with a mean FEV_1_/FVC of 81%. Mean predicted FEV_1_ was 2.5 L/s (88%) and mean FVC was 3.1 L/s (87%). Patients with non-urinary cancers showed a mild but statistically significant reduction in both FEV_1_ and FVC (*p* = 0.010 and 0.006), likely due to thoracic surgeries in lung cancer cases.

### 3.2. Cancer Type Distribution

The distribution of cancer types revealed genitourinary malignancies as most frequent (34%, mainly prostate), followed by breast (14%), colorectal (12%), lung (10%), laryngeal and skin (8% each), intracranial (6%), hematologic and parotid (4% each), and other neoplasms (2%). Different neoplasms (e.g., lung, thyroid, parotid, laryngeal, intracranial, stomach) are exclusively present in the X_0_ group or in the combined cohort, with no representation in X_1_. Skin, breast, and colorectal cancers show overlapping presence in both groups, indicating shared clinical features. The X_1_ group displays a higher relative proportion of skin cancer, possibly reflecting specific pathophysiological or demographic patterns (Figure 1). Interestingly, 10% of patients presented with multiple primary tumors, most commonly prostate and skin basal cell carcinoma, or prostate and colorectal cancer, suggesting a possible oncological susceptibility in this population (Figure 2).

### 3.3. Multivariate Analysis: Factorial Analysis of Mixed Data and Hierarchical Clustering Analysis

FAMD identified a distinct cluster of OSA patients with urinary tract malignancies. Specifically, the first three factorial dimensions delineated a pattern associating genitourinary cancer with specific clinical features (Figure 3). The resulting dendrogram evidenced two principal clusters, which were confirmed by visual inspection of the factor map: Cluster 1 included patients with non-urinary cancer (X_0_), characterized by lower respiratory function, reduced C-PAP compliance, and higher prevalence of pulmonary tumors; Cluster 2 grouped patients with urinary cancer (X_1_), showing better respiratory indices, higher compliance to C-PAP therapy, male sex, and frequent comorbid hypertension. In particular, respiratory function parameters differed significantly between groups, with very large effect sizes. FEV_1_ was significantly higher in the X1 group (3.5 ± 1.0 L/s) compared to X0 (2.3 ± 0.8 L/s, *p* = 0.010, Cohen’s d = 1.38), representing a clinically meaningful difference (Table 1). Similarly, FVC was significantly higher in X1 (4.3 ± 1.0 L/s) versus X0 (2.8 ± 0.9 L/s, *p* = 0.006, Cohen’s d = 1.61). When expressed as percentages of predicted values, FEV_1_% (101.0 ± 9.0% vs. 84.0 ± 21.0%, *p* = 0.100, d = 0.95) and FVC% (100.0 ± 8.0% vs. 84.0 ± 18.0%, *p* = 0.090, d = 1.04) also showed large effect sizes, though these comparisons did not reach statistical significance, likely due to limited statistical power. The FEV_1_/FVC ratio was similar between groups (0.81 in both, *p* = 0.440, d = 0.00) (Table 1). Additional OSA severity parameters showed consistent patterns. Oxygen desaturation index (ODI) was lower in X_1_ (22.0 ± 14.0 events/h) compared to X_0_ (33.0 ± 26.0 events/h, *p* = 0.300, Cohen’s d = −0.48), though this small effect did not reach statistical significance. Similarly, the percentage of time spent with oxygen saturation below 90% (t90) was markedly lower in X_1_ (6.0 ± 10.0%) vs. X_0_ (18.0 ± 25.0%, *p* = 0.100, Cohen’s d = −0.57), representing a medium effect size that approached significance. Mean nocturnal oxygen saturation showed minimal difference between groups (X_1_: 93.0 ± 2.0% vs. X_0_: 94.0 ± 3.0%, *p* = 0.300, Cohen’s d = −0.37). These findings collectively indicate that urinary tract cancer patients in our cohort had less severe nocturnal hypoxemia compared to patients with other cancer types, consistent with their better baseline respiratory function.

Overall, this stratification supports the hypothesis of a distinct clinical phenotype linking OSA to genitourinary malignancies. (Figure 4).

## 4. Discussion

In this retrospective cohort study of 50 patients with both OSA and cancer enrolled in the Vercelli Hospital Registry, we found that urinary tract malignancies (prostate, bladder, and kidney) were the most prevalent cancer type, accounting for 34% (*n* = 17) of cases, followed by breast cancer (16%, *n* = 8), colorectal cancer (12%, *n* = 6), and lung cancer (10%, *n* = 5). Non-urinary tract cancers collectively represented 66% (*n* = 33) of the cohort. The prevalence of urinary tract cancer was significantly higher among male patients (88% in the urinary tract cancer group vs. 58% in the non-urinary cancer group, *p* = 0.043), suggesting a sex-specific vulnerability in OSA-associated malignancies. Furthermore, our analysis revealed two distinct clinical profiles: (1) patients with non-genitourinary cancers (e.g., lung, breast, and colorectal), characterized by lower respiratory function, poorer adherence to C-PAP therapy, and a higher incidence of pulmonary tumors; and (2) patients with genitourinary malignancies, displaying better respiratory indices, greater C-PAP compliance, and frequent hypertension and cardiovascular comorbidities. The urinary tract cancer subgroup was predominantly composed of overweight individuals (mean BMI: 29 kg/m^2^), most of whom were current or former smokers (73%) and had a high prevalence of cardiovascular comorbidities, particularly hypertension (100%). These patterns suggest heterogeneity in cancer distribution among OSA patients and support the hypothesis of pathophysiological mechanisms linking OSA with specific malignancies.

Notably, urinary tract cancer patients demonstrated not only better respiratory function but also less severe OSA phenotype, as evidenced by lower AHI (d = −0.50), ODI (d = −0.48), and t90 (d = −0.57). While these differences did not reach statistical significance due to limited power, the consistent direction of effects across multiple OSA severity markers suggests a distinct clinical phenotype characterized by well-preserved respiratory capacity and milder nocturnal hypoxemia. This pattern may reflect selection bias, as patients with more severe respiratory compromise may have been less likely to survive cancer or may have been excluded due to incomplete data. Alternatively, it may indicate that the OSA-urinary tract cancer association is not primarily driven by hypoxia severity but rather by other mechanisms such as chronic inflammation, immune dysregulation, or shared risk factors (e.g., obesity, metabolic syndrome). The higher C-PAP compliance in this subgroup (94% vs. 61%, *p* = 0.018) may also contribute to better oxygenation parameters, though our cross-sectional design cannot establish temporal relationships

The association between OSA and cancer has garnered increasing attention in recent years. Epidemiological studies suggest that OSA may influence not only cancer incidence but also tumor type and aggressiveness [18,40,41]. However, findings from meta-analyses exploring the association between OSA and overall cancer risk have yielded conflicting results, likely due to differences in OSA severity, diagnostic criteria, and cancer types across studies [42,43]. Mechanistic studies have proposed a unifying immunological hypothesis linking OSA to cancer through immune dysregulation and chronic inflammation [44]. A recent nationwide register-based cohort study found increased cancer risk in OSA patients, particularly for hormone-related and obesity-related malignancies [45]. Recent research has reported a higher risk of kidney and bladder cancer among OSA patients [46], with a systematic review and meta-analysis confirming significant associations between OSA and urological malignancies, particularly in male patients.

Experimental data support intermittent hypoxia—a hallmark of OSA—as a key driver of tumorigenesis. Hypoxia-inducible factors (HIF-1α and HIF-2α) are upregulated under intermittent hypoxic conditions and have been implicated in the progression of bladder, prostate, and renal cancers [47,48,49]. In clear cell renal cell carcinoma, the loss of von Hippel-Lindau tumor suppressor proteins exacerbates HIF accumulation, promoting angiogenesis and tumor growth [50]. Furthermore, intermittent hypoxia stimulates VEGF and NF-κB expression, further enhancing pro-angiogenic and inflammatory pathways [51]. The higher prevalence of urinary tract cancers observed in OSA patients in our study is consistent with this mechanistic hypothesis, suggesting that chronic intermittent hypoxia may act as a biological driver of carcinogenesis in hypoxia-sensitive tissues. Although we did not assess these molecular pathways directly, future studies incorporating HIF expression, VEGF levels, and oxidative stress biomarkers are warranted to determine causality and further elucidate the molecular interplay between OSA and urinary tract malignancies.

The marked male predominance in urinary tract cancer observed in our cohort (88% male in X1 vs. 58% in X0, *p* = 0.043) is consistent with known sex differences in both OSA prevalence and urological cancer incidence. Men have 2–3 times higher OSA prevalence than women, and prostate cancer is exclusively male while bladder and kidney cancers show 3–4-fold higher incidence in men [52]. Furthermore, emerging evidence suggests sex-specific differences in hypoxia response pathways, with males showing greater HIF-1α activation and more pronounced inflammatory responses to intermittent hypoxia [53]. These biological sex differences, combined with higher smoking rates and occupational carcinogen exposures in men, may explain the strong male predominance in OSA-associated urinary tract cancer.

Age is another critical factor in the OSA-cancer relationship. Cancer incidence increases exponentially with age, and OSA prevalence also rises with aging, reaching 30–50% in adults over 65 years [54]. The cumulative exposure to intermittent hypoxia over decades may have a dose-dependent effect on carcinogenesis, though our cross-sectional design cannot assess this temporal relationship. Future studies should stratify analyses by age groups and assess the duration of untreated OSA as a potential risk modifier.

In addition to hypoxia, sleep fragmentation—another core feature of OSA—activates the sympathetic nervous system, leading to increased production of reactive oxygen species (ROS) and immune dysregulation [55,56,57]. Recent findings by Norouzi et al. have shown that disruption of slow-wave sleep (SWS), a restorative phase of the sleep cycle, is significantly associated with lower blood SpO_2_ in OSA patients. This disruption not only reflects sleep fragmentation but also contributes to heightened sympathetic nervous system activity, increased oxidative stress, and inflammatory signaling, all of which may create a permissive tumor microenvironment, particularly in hypoxia-sensitive tissues, such as the urinary tract [58]. These mechanisms collectively contribute to chronic inflammation, impaired immune surveillance, and enhanced tumor proliferation and angiogenesis [59,60].

Our findings align with previous reports suggesting that urinary tract tissues are particularly sensitive to intermittent hypoxia and oxygen desaturation during sleep [61]. Repetitive airway obstruction and nocturnal hypoxia may trigger systemic pathophysiological responses that promote carcinogenesis. Several studies have also linked short sleep duration and poor sleep quality to increased cancer risk [3,57,62]. Recent meta-analysis confirmed a correlation between OSA severity and cancer incidence, reinforcing the hypothesis that chronic intermittent hypoxia contributes to tumor initiation and progression [57,62].

In our cohort, breast (14%) and colorectal cancer (12%) were also frequently observed. These findings are consistent with data from the ESADA study, which reported a higher prevalence of breast and urinary tract cancers in OSA patients [3]. However, a notable divergence emerged in sex distribution: while ESADA found increased cancer prevalence in European females with OSA and nocturnal hypoxia, our registry data show a predominance of urinary tract cancers in men. Such discrepancy may de ascribable to population-specific differences or methodological variations in OSA assessment.

Interestingly, 10% of OSA patients with cancer in our cohort presented with multiple primary neoplasms, such as prostate and colorectal cancer or prostate and skin basal cell carcinoma. This observation raises the possibility of a broader oncological vulnerability in OSA patients, particularly those with genitourinary involvement.

Furthermore, our analysis shows two distinct clinical profiles: (1) patients with non-genitourinary cancers (e.g., lung, breast, and colorectal), characterized by lower respiratory function, poorer adherence to C-PAP therapy, and a higher incidence of pulmonary tumors; and (2) patients with genitourinary malignancies, displaying better respiratory indices, greater C-PAP compliance, and frequent hypertension and cardiovascular comorbidities. These patterns suggest heterogeneity in cancer distribution among OSA patients and support the hypothesis of pathophysiological mechanisms linking OSA with specific malignancies.

The complex comorbidity profiles observed in our cohort may explain the bidirectional role of OSA in systemic disease [63]. Recent evidence examining the relationship between chronic insomnia and obstructive sleep apnea, two conditions that frequently coexist and may interact pathophysiologically giving rise to a disease known as COMISA, suggests that the combined burden of fragmented sleep and chronic nocturnal hypoxia may exert synergistic systemic stress, potentially exacerbating oncological susceptibility. As highlighted by Khazaie et al., the coexistence of insomnia and OSA represents a more complex sleep disorder phenotype, which may influence cancer risk through mechanisms extending beyond intermittent hypoxia alone [64].

Taken together, our findings support the hypothesis of a bidirectional relationship between OSA and cancer, with intermittent hypoxia and sleep fragmentation acting as potential mediators of tumorigenesis. Further studies are warranted to clarify the impact of OSA on cancer incidence, progression, and outcomes, particularly in genitourinary malignancies, and to determine whether PAP therapy mitigates these risks [3,65]. Understanding the differential impact of OSA on specific cancer types may inform targeted surveillance strategies and improve oncologic risk stratification.

The main limitation of the present study is the relatively small sample size of the group which may reduce the statistical power and generalizability of the findings. Adjustment for important confounders of cancer risk, such as genetic predisposition, physical activity levels, occupational exposures, and dietary habits, was not possible. Moreover, we did not observe a statistically significant association between OSA and cancer in female patients. This contrasts with findings from larger cohorts, such as the ESADA study, which reported a higher cancer prevalence in women with OSA and nocturnal hypoxia. This discrepancy may be explained by several factors, including cancer subtype distribution, hormonal influences on tumor biology and immune modulation, gender-specific patterns of smoking and environmental exposure, which were not evaluated. In addition, our study did not assess cancer incidence rates, limiting our ability to determine whether OSA is associated with an increased risk of developing cancer over time. Longitudinal studies are needed to clarify whether OSA contributes to cancer onset, progression, or mortality, particularly in relation to genitourinary neoplasm. It should also be acknowledged that the two subgroups—patients with urinary vs. non-urinary tract cancers—were unbalanced in size (33 vs. 17), reflecting the real-world distribution of malignancies in the registry. This heterogeneity may limit baseline comparability and reduce statistical power for between-group analyses. Moreover, we did not measure hypoxia-related biomarkers or molecular mechanisms (HIF, VEGF, NF-κB), limiting our ability to infer causation. The discussion of these pathways is based on existing literature and represents potential mechanisms that warrant further investigation. Finally, we did not perform multivariable adjustments for potential confounders such as age, sex, smoking status, obesity, or hypertension, which are known to influence both OSA and cancer risk. Given the retrospective and exploratory design, our analyses were primarily descriptive and intended to identify potential clinical patterns. Therefore, the results should not be interpreted as causal but rather as hypothesis-generating, warranting further validation in larger prospective studies using adjusted models.

Future research should aim to include larger, gender-balanced cohorts, incorporate prospective designs, and adjust for lifestyle and genetic factors to better elucidate the potential causal relationship between OSA and cancer incidence.

## 5. Conclusions

Our results are consistent with a higher prevalence of urinary tract and breast cancer in OSA patients. In particular, urinary tract cancers were significantly more frequent in male patients and associated with distinctive clinical profiles characterized by preserved respiratory indices, coexisting hypertension, and higher adherence to C-PAP therapy. These results support further investigation into potential mechanistic links among sleep apnea, nocturnal hypoxia, sleep fragmentation, and urinary tract carcinogenesis. Such associations could have therapeutic implications particularly regarding the protective role of PAP therapy. Large-scale, prospective studies are needed to determine whether OSA contributes causally to cancer onset or progression and whether effective OSA treatment modifies cancer risk and outcomes.

## Figures and Tables

**Figure 1 arm-93-00054-f001:**
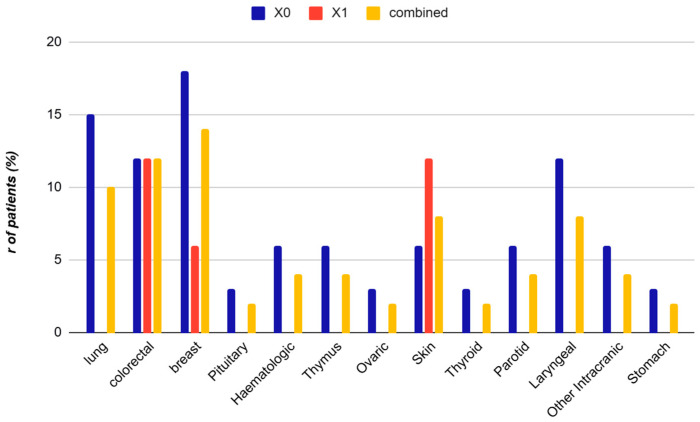
Distribution of cancer types among OSA patients according to subgroup. Different neoplasms (e.g., lung, thyroid, parotid, laryngeal, intracranial, stomach) are exclusively present in the X_0_ group (non-urinary tract cancer) or in the combined cohort, with no representation in X_1_ (urinary tract cancer), suggesting distinct clinical distributions. Skin, breast, and colorectal cancers show overlapping presence in both groups, indicating shared clinical features. The X_1_ group displays a higher relative proportion of skin cancer, possibly reflecting specific pathophysiological or demographic patterns. Data are expressed as percentage of patients (%). Abbreviations: X_0_ = non-urinary tract cancer; X_1_ = urinary tract cancer.

**Figure 2 arm-93-00054-f002:**
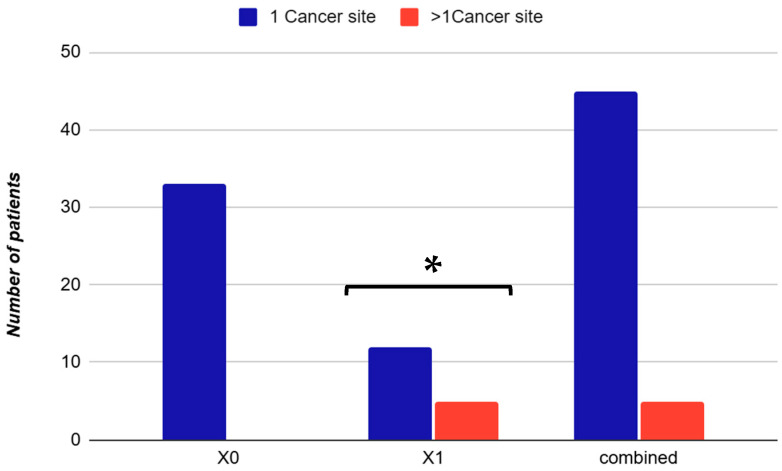
Distribution of single and multiple cancer sites in OSA patients by cancer subgroup. Proportion of patients with a single cancer site (blue) vs. multiple cancer sites (orange) across three categories: non-genitourinary cancers (X_0_), genitourinary cancers (X_1_), and the overall cohort. In the overall cohort, single-site cancers accounted 45/50 patients, (~90%), while multiple-site cancers included 5/50 patients (10%). Only patients in the X1 group present with a significant proportion of multiple neoplasms (5/17 patients, ~29.4%), suggesting a possible oncological vulnerability in OSA patients with genitourinary malignancies (*p* = 0.001). Data are expressed as *n*. of patients. Abbreviations: X_0_ = non-urinary tract cancer; X_1_ = urinary tract cancer. * Statistical significance: *p* < 0.05.

**Figure 3 arm-93-00054-f003:**
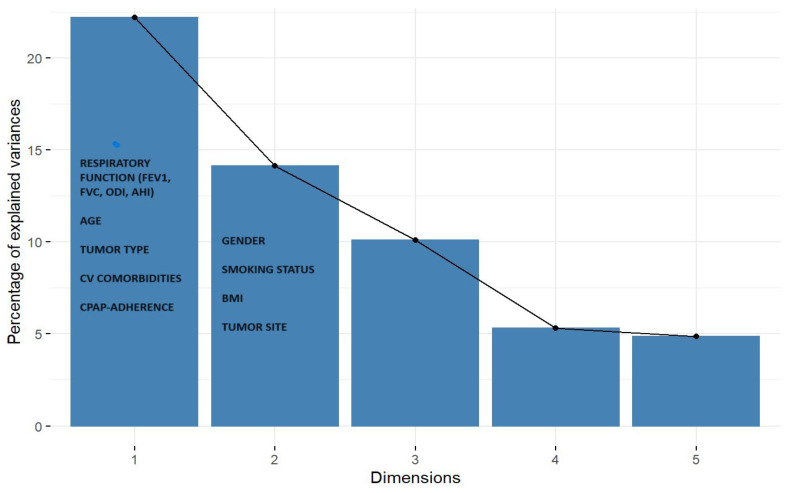
Scree plot of the FAMD analysis. The graph shows the distribution of the variance explained by each component. The first three dimensions together account for 46.43% of the total variability. Dimension 1 is strongly influenced by respiratory variables (i.e., FEV_1_, FVC, AHI, ODI) and age, while dimension 2 is associated with sex, smoking status, and BMI. Patients with genitourinary cancer (X_1_) are prominently represented along Dim.1 and Dim.2. Abbreviations: AHI = apnea–hypopnea index; ODI = oxygen desaturation index; FEV_1_ = forced expiratory volume in one second; FVC = forced vital capacity; X_1_ = urinary tract cancer.

**Figure 4 arm-93-00054-f004:**
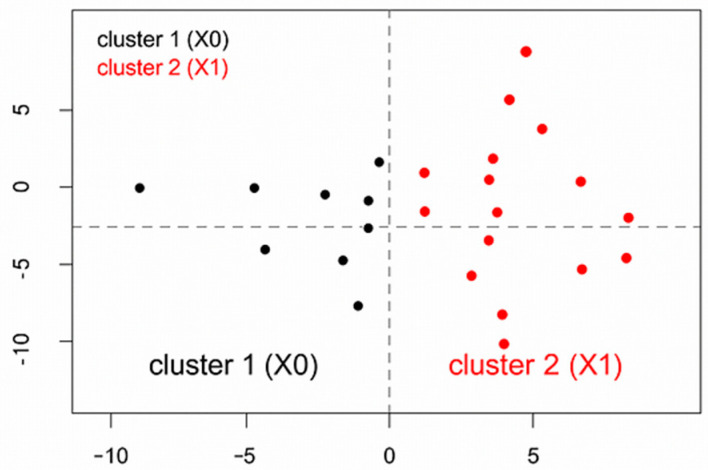
Factor map of OSA patients distributed according to the first two dimensions of the FAMD. The plot displays individual data points based on factorial coordinates. The black points (Cluster 1) represent patients without urinary cancer (X_0_), while red points (Cluster 2) correspond to those with urinary cancer (X_1_). The clear separation between clusters indicates distinct clinical profiles: Cluster 1 (X_0_) is associated with lower respiratory function, reduced C-PAP adherence, and a higher prevalence of thoracic or pulmonary tumors, whereas Cluster 2 is characterized by hypertension, male predominance, greater compliance with C-PAP therapy, and preserved respiratory indices.

**Table 1 arm-93-00054-t001:** Main clinical characteristics of the whole study cohort and baseline descriptive statistics.

Variable	Non-Urinary (X0) (*n* = 33)	Urinary (X1) (*n* = 17)	Combined (*n* = 50)	Effect Size (Cohen’s d/h)	OR (CI 95%)	*p*-Value
Male sex, *n* (%)	20 (61%)	15 (88%)	35 (70%)	h = 0.66	0.21 [0.04; 1.05]	0.043 *
Age (years)	68.0 ± 7.0	65.0 ± 3.0	67.0 ± 7.0	d = −0.50	0.94 [0.87; 1.02]	0.107
BMI (kg/m^2^)	29.0 ± 3.0	29.0 ± 4.0	29.0 ± 4.0	d = 0.00	1.01 [0.85; 1.20]	0.958
Smoking habit, *n* (%)	25 (76%)	12 (73%)	37 (74%)	h = −0.12	0.83 [0.16; 4.40]	0.830
Allergy, *n* (%)	11 (32%)	5 (30%)	16 (32%)	h = −0.09	0.93 [0.18; 4.90]	0.900
CV Disease, *n* (%)	28 (85%)	17 (100%)	45 (90%)	h = 0.80	NE	0.300
C-PAP treatment, *n* (%)	25 (76%)	13 (77%)	38 (76%)	h = 0.02	0.95 [0.20; 4.63]	0.900
C-PAP compliance, *n* (%)	20 (61%)	16 (94%)	36 (72%)	h = 0.87	8.98 [1.50; 236]	0.018 *
AHI (events/h)	37.0 ± 30.0	24.0 ± 15.0	32.0 ± 26.0	d = −0.50	0.97 [0.94; 1.01]	0.200
Mean SpO_2_ (%)	94.0 ± 3.0	93.0 ± 2.0	93.0 ± 2.0	d = −0.37	0.81 [0.56; 1.17]	0.300
FEV_1_ (L/s)	2.3 ± 0.8	3.5 ± 1.0	2.5 ± 1.0	d = 1.38	4.22 [0.88; 20.1]	0.010 **
FEV_1_ (%)	84.0 ± 21.0	101.0 ± 9.0	88.0 ± 21.0	d = 0.95	1.07 [0.97; 1.18]	0.100
FVC (L/s)	2.8 ± 0.9	4.3 ± 1.0	3.1 ± 1.1	d = 1.61	4.05 [0.96; 17.1]	0.006 **
FVC (%)	84.0 ± 18.0	100.0 ± 8.0	87.0 ± 17.0	d = 1.04	1.08 [0.98; 1.18]	0.090
FEV_1_/FVC	0.81 ± 0.11	0.81 ± 0.05	0.81 ± 0.09	d = 0.00	0.96 [0.00; 1774]	0.440
ODI	33.0 ± 26.0	22.0 ± 14.0	29.0 ± 23.0	d = −0.48	0.97 [0.93; 1.02]	0.300
t90 (%)	18.0 ± 25.0	6.0 ± 10.0	14.0 ± 21.0	d = −0.57	0.95 [0.88; 1.03]	0.100
Mean SpO_2_ (%)	94.0 ± 3.0	93.0 ± 2.0	93.0 ± 2.0	d = −0.37	0.81 [0.56; 1.17]	0.300

Data are presented as mean ± standard deviation for continuous variables and *n* (%) for categorical variables. Effect sizes are reported as Cohen’s d for continuous variables and Cohen’s h for categorical variables. Effect size interpretation: negligible (|d| or |h| < 0.2), small (0.2–0.5), medium (0.5–0.8), large (≥0.8). Positive values indicate higher values in the urinary tract cancer group (X1) compared to the non-urinary tract cancer group (X0); negative values indicate lower values in X1 compared to X0. OR = odds ratio; CI = confidence interval; CV = cardiovascular; C-PAP = continuous positive airway pressure; AHI = apnea-hypopnea index; ODI = oxygen desaturation index; t90 = percentage of time with SpO_2_ < 90%; SpO_2_ = oxygen saturation; FEV_1_ = forced expiratory volume in 1 s; FVC = forced vital capacity; NE = not estimable. * *p* < 0.05; ** *p* < 0.01.

## Data Availability

The raw data supporting the conclusions of this article will be made available by the authors on request.

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
