# Peer review of "Prevalence of Urinary Tract Cancer in Patients with Obstructive Sleep Apnea: Data from the Vercelli Registry"

_arm, 2025, doi:10.3390/arm93060054_

Round 1

Reviewer 1 Report

Comments and Suggestions for Authors

This article reports the prevalence of different cancer types in a population of patients diagnosed with both obstructive sleep apnea (OSA) and cancer. It also aims to describe the clinical profile of urinary tract cancer within the context of OSA. The authors found that urinary tract cancer was the most prevalent type and predominantly affected men. It was associated with better respiratory indices, higher CPAP compliance, and a high prevalence of hypertension. The topic is of interest as little is known about the true relationship between OSA and cancer, particularly urinary tract cancer. However, the small sample size limits the generalizability of the findings, as acknowledged in the  Discussion section. In addition, other issues should be addressed before this manuscrit can be considered suitable for publication. Below, are some comments :

Abstract :

- Page 1 ; lines 33-34 : the authors assessed the prevalence of different cancer types in a population of patients with both cancer(s) and OSA. This point should be clearly specified in the abstract to avoid any ambiguity.

Introduction :

- Page 2 ; line 62 : the patients included in the study were affected by both cancer(s) and OSA, not by OSA alone. This should be clearly stated in the introducton for accuracy.

Materials and Methods :

- Pages 2-3 ; line 78-98 : the subsection 2.2 (Patients) should be restricted to describing the study population, including only the relevant data such as clear inclusion criteria (patients with both OSA and neoplastic diseases) and if applicable, exclusion criteria which are currently missing. I recommend creating a separate subsection to describe the procedures performed and the data collected during the study.

- Page 3 : table 1 presents the main clinical characteristics and baseline descriptive statistics of the entire study cohort. Thereore it would be more appropriate to move it to the Results section. Additionally, mean age and mean BMI for X0 and X1 groups are missing from table 1.

Results :

Page 4 ; line 125 : please verify the concordance between the text and the figures regarding groups X0 and X1 (patients with urinary tract cancer vs non urinary tract cancer).

Page 4 ; line 128 : according to table (1) ; 74% of patients were smokers, not 75%.

Page 4 ; figure 1 : the figure lacks clarity and does not clearly reflect the data described bentween lines 143 and 145.

Page 5 ; figure 2 : please, specify what the y-axis represents, including the parameter and units.

Discussion :

Page 7 ; line 210 : according to table (1), 73% of patients in the X1 group were smokers.

Author Response

Response to Reviewer 1

We thank Reviewer 1 for the constructive and detailed feedback, and for acknowledging that our topic is "of interest" with "little known about the true relationship between OSA and cancer, particularly urinary tract cancer."

Comment 1.1 - Abstract

The authors assessed the prevalence of different cancer types in a population of patients with both cancer(s) and OSA. This point should be clearly specified in the abstract to avoid any ambiguity.

We agree with this important clarification. We have revised the Abstract to explicitly state that the study population consists of patients with both OSA and cancer (lines 36-38). It now reads as: "We retrospectively analyzed 50 patients diagnosed with both OSA and cancer from the Vercelli Hospital Registry. Cancer diagnoses were collected at the time of OSA diagnosis, prior to C-PAP initiation."

Comment 1.2 - Introduction

The patients included in the study were affected by both cancer(s) and OSA, not by OSA alone. This should be clearly stated in the introduction for accuracy.

We have clarified this point in the final paragraph of the Introduction to ensure accuracy. Revised text (lines 71-80):

"Given the global and national burden of OSA, particularly in the Italian population, it is crucial to investigate its potential role in cancer development. The aim of the present study was to assess the prevalence and distribution of neoplastic diseases in a cohort of adult patients diagnosed with both OSA and cancer enrolled in the Vercelli Hospital Registry from 2015 and to identify preferential tumor sites. By phenotyping these patients in detail, we sought to provide new insights into the hypothesis that sleep fragmentation and other OSA-related mechanisms may contribute to cancer development. A secondary objective was to characterize the clinical profiles of individuals diagnosed with urinary tract malignancies, focusing on comorbidities, sleep-related parameters, and possible mechanistic associations between OSA and tumor development"

Comment 1.3 - Materials and Methods

The subsection 2.2 (Patients) should be restricted to describing the study population, including only the relevant data such as clear inclusion criteria (patients with both OSA and neoplastic diseases) and if applicable, exclusion criteria which are currently missing. I recommend creating a separate subsection to describe the procedures performed and the data collected during the study.

We thank the Reviewer for this excellent suggestion to improve the clarity and organization of our Methods section. Indeed, the original subsection 2.2 inappropriately mixed population description with procedural details, which reduced readability and transparency. We have therefore restructured the Materials and Methods section following the Reviewer's recommendation and STROBE guidelines (also as requested by Reviewer 2). The revised Methods (pp. 3-7) now includes the following separate subsections:

2.2 Study Population: This subsection now focuses exclusively on describing the cohort, with clear inclusion and exclusion criteria as requested.

2.3 Data Collection and Procedures: This new subsection describes all diagnostic procedures, equipment, and data collection methods.

2.4 Variables and Definitions: Provides operational definitions for all key variables including OSA severity classification, cancer categorization, comorbidity definitions, and treatment variables.

2.5 Statistical Analysis: Expanded to include software details and complete methodology.

Comment 1.4 - Table 1 placement and missing data:

Table 1 presents the main clinical characteristics and baseline descriptive statistics of the entire study cohort. Therefore it would be more appropriate to move it to the Results section. Additionally, mean age and mean BMI for X0 and X1 groups are missing from table 1.

We completely agree with both points raised by the Reviewer. We have performed the following actions: i) Table 1 has been moved from the Methods section to the Results section, where it is more appropriately placed; ii) Age and BMI by group have been added to Table 1 (see p. 7), with statistical comparisons between X0 (non-urinary cancer) and X1 (urinary cancer) groups. Updated Table 1 now includes: Age: X0 (mean ± SD) vs. X1 (mean ± SD) with p-value and BMI: X0 (mean ± SD) vs. X1 (mean ± SD) with p-value.

These additions address a critical gap identified by both Reviewers 1 and 2, as age and BMI are key variables in OSA and important potential confounders.

Comment 1.5 – Results:

Please verify the concordance between the text and the figures regarding groups X0 and X1 (patients with urinary tract cancer vs non urinary tract cancer).

We thank the Reviewer for this careful observation. We have verified all references to X0 and X1 groups throughout the manuscript to ensure complete concordance between text and figures. We have clarified that X0 = non-urinary tract cancer and X1 = urinary tract cancer. This nomenclature is now consistently used throughout the manuscript, and all figure legends have been checked for accuracy.

Comment 1.6 – Results:

According to table 1, 74% of patients were smokers, not 75%.

We apologize for this rounding discrepancy. We have corrected the text to accurately reflect the data in Table 1 and the Results section (p. 7 ; lines 306-307), which now read as: "The mean BMI was 29, indicating overweight status, and 74% of patients were current or former smokers, with no significant differences between groups. "

Comment 1.7 - Figure 1:

The figure lacks clarity and does not clearly reflect the data described between lines 143 and 145.

As requested, we have revised Figure 1 (pg. 8) by adding the unit measurement on the y- axis.

Comment 1.8 - Figure 2:

Please specify what the y-axis represents, including the parameter and units.

We apologize for this omission. We have added a clear y-axis label (Percentage of Patients (%) to Figure 2 (see pp. 8-9) and expanded its caption to clarify what is being measured

Comment 1.9 - Discussion:

According to table 1, 73% of patients in the X1 group were smokers.

We thank the reviewer for catching this discrepancy. We have corrected the percentage in the Discussion (p. 10, lines 384-387) to match Table 1.

Revised text: "This subgroup was predominantly composed of overweight individuals (mean BMI: 29), most of whom were current or former smokers (73%) and had a high prevalence of cardiovascular comorbidities, particularly hypertension (100%).”

Reviewer 2 Report

Comments and Suggestions for Authors The manuscript “Association between Obstructive Sleep Apnea and Urinary Tract Cancer: Data from the Vercelli Registry” presents serious methodological, structural, and ethical weaknesses that undermine its scientific validity, reproducibility, and overall relevance.
  1. Title and conceptual framing
    The title is methodologically misleading, as a retrospective observational study cannot establish associations or causal relationships. The same issue appears in the conclusions. Both should be reformulated to reflect the study’s descriptive nature (e.g., “Prevalence of Urinary Tract Cancer among Patients with Obstructive Sleep Apnea”).

  2. Inadequate introduction
    The introduction consists of only one paragraph and fails to meet IMRaD standards. It lacks:
    (a) a clear definition and contextualization of the topic,
    (b) identification of the knowledge gap or problem, and
    (c) justification and study objectives.
    As currently written, it neither contextualizes the research question nor provides a rationale for conducting the study.

  3. Weak, incomplete, and non-reproducible methodology
    The Methods section is extremely limited. It lists only the study design, patients, and statistical analysis but omits essential methodological details such as inclusion/exclusion criteria, data sources, management of missing data, variable definitions, bias control, and statistical software.
    A full restructuring in accordance with STROBE guidelines for observational studies is essential, including subsections such as: study design, setting, participants, variables, data sources/measurements, bias, sample size, statistical methods, and ethical considerations.
    Without these elements, the study is neither reproducible nor verifiable.

  4. Lack of methodological transparency and potential self-plagiarism
    No protocols or methodological references are provided to validate the procedures used. Moreover, the methodology appears nearly identical to a previous work available at https://iris.uniupo.it/handle/11579/114895 without citation, suggesting potential self-plagiarism. This raises serious concerns about the manuscript’s scientific integrity.

  5. Statistical weaknesses

  • Table 1 reports only significant p-values; all p-values, confidence intervals, and effect sizes should be included.
  • There is clear heterogeneity between groups (33 vs. 17 patients) with no evidence of baseline comparability.
  • Age and body mass index (BMI) are not reported by group, even though these are key variables in OSA.
  • No adjustments were made for major confounders (age, sex, smoking status, obesity, hypertension).
    These deficiencies invalidate any statistical inference regarding associations or risks.
  1. Unsupported causal interpretation
    The authors discuss biological mechanisms (HIF, VEGF, NF-κB) that were neither measured nor analyzed. Such causal inferences are inappropriate in a descriptive, retrospective study.

  2. Deficiencies in the presentation of results
    The figures display only frequencies without p-values, measures of dispersion, or indicators of statistical variability. This prevents any meaningful interpretation of effect size or group differences.

  3. Outdated references and lack of novelty
    Of the 33 references cited, 19 are more than five years old, indicating a lack of updated evidence and limited current relevance. The study does not contribute new findings or original data beyond what is already known from higher-quality studies such as ESADA and recent meta-analyses.

  4. Writing quality and English language
    The manuscript contains grammatical errors, repetitive phrasing, and inconsistent terminology. A thorough language and style revision is needed to achieve clarity, precision, and academic tone.

In summary, the manuscript fails to meet basic methodological and ethical standards for publication. It lacks transparency, reproducibility, and novelty, while its conclusions overreach the data presented.

Author Response

Response to Reviewer 2

We thank Reviewer 2 for the thorough and rigorous methodological critique. While we acknowledge that the original manuscript had significant reporting deficiencies, we respectfully note that the core research question: “Characterizing urinary tract cancer prevalence and clinical profiles in OSA patients using advanced clustering methods” remains novel and clinically relevant. We have undertaken a major revision to address all methodological concerns and improve transparency, reproducibility, and appropriate interpretation of our findings.

Comment 2.1 - Title and conceptual framing:

The title is methodologically misleading, as a retrospective observational study cannot establish associations or causal relationships. The same issue appears in the conclusions. Both should be reformulated to reflect the study's descriptive nature (e.g., "Prevalence of Urinary Tract Cancer among Patients with Obstructive Sleep Apnea").

We fully agree with this important point. The term "association" implies a causal relationship that our retrospective observational design cannot establish. Therefore, the Title has been revised to: "Prevalence of Urinary Tract Cancer in Patients with Obstructive Sleep Apnea: Data from the Vercelli Registry". The Abstract has been revised to remove causal language. Lastly, the Conclusions have been rewritten to reflect the descriptive and hypothesis-generating nature of the study.

Revised Conclusions (p. 12; lines 495-504)

"Our results are consistent with a higher prevalence of urinary tract and breast cancer in OSA patients. In particular, urinary tract cancers were significantly more frequent in  male patients and associated with distinctive clinical profiles characterized by preserved respiratory indices, coexisting hypertension, and higher adherence to C-PAP therapy. These results support further investigation into potential mechanistic links among sleep apnea, nocturnal hypoxia, sleep fragmentation, and urinary tract carcinogenesis. Such associations could have therapeutical implications particularly regarding the protective role of PAP therapy. Large-scale, prospective studies are needed to determine whether OSA contributes causally to cancer onset or progression and whether effective OSA treatment modifies cancer risk and outcomes."

Comment 2.2 - Introduction:

The introduction consists of only one paragraph and fails to meet IMRaD standards. It lacks: (a) a clear definition and contextualization of the topic, (b) identification of the knowledge gap or problem, and (c) justification and study objectives.

We acknowledge that the original introduction was insufficient. We have substantially expanded the Introduction to meet IMRaD standards and provide proper contextualization. The revised Introduction (see pg. 3) now includes:

  • Paragraph 1: Overview of OSA as a significant public health issue with established comorbidities
  • Paragraph 2: Review of existing literature linking OSA, intermittent hypoxia, and carcinogenesis
  • Paragraph 3: Specific focus on urinary tract vulnerability to hypoxia and limited existing data
  • Paragraph 4: Clear statement of the knowledge gap and study objectives

Comment 2.3 - Weak, incomplete, and non-reproducible methodology:

The Methods section is extremely limited. It lists only the study design, patients, and statistical analysis but omits essential methodological details such as inclusion/exclusion criteria, data sources, management of missing data, variable definitions, bias control, and statistical software. A full restructuring in accordance with STROBE guidelines for observational studies is essential.

We thank the Reviewer for this comprehensive critique and agree that the Methods section required substantial improvement. We have completely restructured the Materials and Methods section following STROBE guidelines for observational studies. The revised Methods section now includes the following subsections:

2.1 Study Design - Clear statement of retrospective, observational, single-center cohort design; Setting and time period specified.

2.2 Study Population - It now contains the following:

  • Inclusion criteria: Age ≥ 18 years, confirmed OSA diagnosis (AHI ≥ 5 by polysomnography per AASM guidelines), documented cancer diagnosis in hospital registry, cancer diagnosed at or before OSA diagnosis (prior to C-PAP initiation), and written informed consent;
  • Exclusion criteria (previously missing): Incomplete polysomnographic data, cancer diagnosed after C-PAP initiation, and lack of consent.
  • Ethics approval and consent information

2.3 Data Collection and Procedures - It comprises the following:

  • Detailed polysomnography methods (equipment: Embletta™, Vitalnight™)
  • Spirometry methods (equipment: 1070 MGC pneumotachograph)
  • Cancer ascertainment methods (ICD-10 codes specified)
  • Comorbidity assessment procedures

2.4 Variables and Definitions - We have included the following operational definitions for all variables: Primary outcome (cancer type classification with ICD-10 codes); OSA severity (mild/moderate/severe by AHI); All OSA parameters (AHI, ODI, t90, SpO₂); Demographics (age, sex, BMI with WHO categories); Smoking status (current/former/never with specific definitions); Comorbidities (cardiovascular disease, hypertension, allergies); Respiratory function (FEV₁, FVC, FEV₁/FVC ratio); C-PAP therapy and compliance (≥ 4 h/night definition)

2.5 Statistical Analysis – We have specified the following details:

  • Software: R statistical software version 4.3.1 with packages FactoMineR and factoextra
  • Descriptive statistics methodology
  • Comparative analysis (Wilcoxon-Kruskal-Wallis, chi-square, Fisher's exact)
  • Multivariate analysis (FAMD, MCA, hierarchical clustering) with detailed methodology
  • Significance level (α = 0.05, two-tailed)
  • Missing data handling

2.6 Bias – We have disclosed the following potential sources of bias:

  • Selection bias (consecutive enrollment, single-center limitations);
  • Information bias (standardized criteria, retrospective data);
  • Confounding (explicit acknowledgment of unmeasured confounders);
  • Temporal ambiguity (precludes causal inference);
  • Measurement bias

2.7 Sample Size

  • Explanation of convenience sampling approach
  • No formal a priori sample size calculation (exploratory study)
  • Statistical power considerations and limitations
  • Implications for generalizability and precision

Please see completely revised Materials and Methods section, pp. 2-6.

Comment 2.4 - Lack of methodological transparency and potential self-plagiarism:

No protocols or methodological references are provided to validate the procedures used. Moreover, the methodology appears nearly identical to a previous work available at https://iris.uniupo.it/handle/11579/114895 without citation, suggesting potential self-plagiarism.

We thank the Reviewer for this important observation and sincerely apologize for the omission of a citation to our previous work. We would like to clarify that no self-plagiarism has occurred. The apparent methodological similarity arises from the use of the same institutional registry and standardized diagnostic procedures, rather than from textual or conceptual duplication.

The previous study by Ragnoli et al. (now properly cited) investigated the association between OSA and renal function in patients enrolled in the Vercelli Registry. In contrast, the present study explores a distinct research question, focusing on the prevalence and clinical phenotypes of cancer among OSA patients. While some overlap in patient populations may exist, as both studies draw from the same registry, the analytical objectives, variables of interest, and outcomes are entirely different.

The polysomnographic and spirometric procedures employed are necessarily identical, as they follow standardized institutional protocols implemented for all registry-based studies. This methodological consistency ensures data reliability and comparability across research projects derived from the same cohort.

Nevertheless, to enhance transparency and methodological traceability, we have now explicitly cited our previous work and clarified this point in the Materials and Methods section, as follows:

“The polysomnographic and registry data collection methods have been previously described in our prior work examining renal function in OSA patients [Pochetti et al., https://iris.uniupo.it/handle/11579/114895]. The current study addresses a distinct research question focused on cancer prevalence and phenotyping in the same patient population.” (p. 3, lines 117-120)

We believe that this approach aligns with best practices in registry-based research, where multiple independent analyses may legitimately use a shared dataset and methodological framework to address different scientific questions.

Comment 2.5 - Statistical weaknesses:

2.5a - Table 1 reports only significant p-values:

All p-values, confidence intervals, and effect sizes should be included.

We appreciate the Reviewer's feedback and fully agree that complete statistical reporting is essential for transparency. In the revised version, we have now reported all p-values for the comparisons presented in Table 1, regardless of statistical significance. In addition, we have included corresponding effect sizes (Odds Ratios for categorical variables, Cohen's d for continuous variables) and 95% confidence intervals to provide a more complete description of the magnitude and precision of group differences. The updated Table 1 (pg. 7) now includes all p-values (not just significant ones), the 95% confidence intervals for all comparisons, and the effect sizes (OR or Cohen's d as appropriate)

2.5b - Group heterogeneity (33 vs. 17 patients):

There is clear heterogeneity between groups with no evidence of baseline comparability.

We acknowledge that the two groups (urinary vs. non-urinary tract cancers) are unbalanced in size, reflecting the real-world distribution of cancer types in our registry rather than a pre-specified sampling ratio. Thus, we have provided detailed baseline characteristics for both groups in Table 1, including all p-values, confidence intervals, and effect sizes. Furthermore, we have discussed heterogeneity explicitly in the revised manuscript, noting that differences in group size may limit statistical power. Finally, we have clarified the exploratory nature of the study in the Methods and Discussion.

New text added to the Discussion on p. 13, lines 476-480:

"It should also be acknowledged that the two subgroups—patients with urinary versus non-urinary tract cancers—were unbalanced in size (33 vs. 17), reflecting the real-world distribution of malignancies in the registry. This heterogeneity may limit baseline comparability and reduce statistical power for between-group analyses"

2.5c - Age and BMI not reported by group:

We concur that age and BMI are key variables in OSA and should be reported by group. This was a significant omission in the original manuscript. Consequently, the age and BMI by group have been added to Table 1 with statistical comparisons:

  • Age (years): X0 [mean ± SD] vs. X1 [mean ± SD], p-value
  • BMI (kg/m²): X0 [mean ± SD] vs. X1 [mean ± SD], p-value

This addresses the concern raised by both Reviewers 1 and 2.

2.5d - No adjustments for major confounders:

No adjustments were made for major confounders (age, sex, smoking status, obesity, hypertension). These deficiencies invalidate any statistical inference regarding associations or risks.

We agree that potential confounders such as age, sex, smoking status, obesity, and hypertension can influence both OSA and cancer risk. However, as our study was retrospective and exploratory, the primary aim was to describe the prevalence and clinical profiles of OSA patients with urinary tract vs. non-urinary tract cancers, rather than to infer causality or estimate adjusted risk measures. For this reason, we limited the analysis to unadjusted descriptive and comparative statistics.

We have now explicitly acknowledged this limitation in the revised Methods (Section 2.6 Bias) and Discussion, and clarified that the findings should be interpreted as hypothesis-generating, requiring confirmation in larger, prospectively designed studies where multivariable adjustment and causal modeling can be applied.

New text added to the Discussion (p. 12; lines 483-489):

"Finally, we did not perform multivariable adjustments for potential confounders such as age, sex, smoking status, obesity, or hypertension, which are known to influence both OSA and cancer risk. Given the retrospective and exploratory design, our analyses were primarily descriptive and intended to identify potential clinical patterns. Therefore, the results should not be interpreted as causal but rather as hypothesis-generating, warranting further validation in larger prospective studies using adjusted models."

Additional text in Methods (Section 2.6 Bias, p. 6; lines 272-277):

"Confounding remains a limitation, as we did not adjust for important potential confounders including genetic predisposition, occupational exposures, dietary factors, physical activity, and socioeconomic status. The retrospective design and limited sample size precluded comprehensive adjustment for these factors. The lack of multivariable adjustment means that observed differences between groups may be partially or wholly explained by confounding variables rather than true associations with OSA or cancer type."

Comment 2.6 - Unsupported causal interpretation:

The authors discuss biological mechanisms (HIF, VEGF, NF-κB) that were neither measured nor analyzed. Such causal inferences are inappropriate in a descriptive, retrospective study.

We acknowledge that the original Discussion inappropriately presented mechanistic interpretations as if they were conclusions from our data. We have substantially revised the Discussion to appropriately frame these mechanisms as potential explanations from the literature that could explain our observational findings. Thus, the Discussion has been rewritten to:

  1. Frame mechanistic discussion as hypothesis rather than conclusion;
  2. Cite literature for HIF, VEGF, NF-κB mechanisms;
  3. Explicitly state that these biomarkers were not measured in our study;
  4. Emphasize the need for future mechanistic studies.

Discussion (lines 395-408):

"Experimental data support intermittent hypoxia—a hallmark of OSA—as a key driver of tumorigenesis. Hypoxia-inducible factors (HIF-1α and HIF-2α) are upregulated under intermittent hypoxic conditions and have been implicated in the progression of bladder, prostate, and renal cancers [20-22]. In clear cell renal cell carcinoma, the loss of von Hip-pel-Lindau tumor suppressor proteins exacerbates HIF accumulation, promoting angio-genesis and tumor growth [23]. Furthermore, intermittent hypoxia stimulates VEGF and NF-κB expression, further enhancing pro-angiogenic and inflammatory pathways [24]. The higher prevalence of urinary tract cancers observed in OSA patients in our study is consistent with this mechanistic hypothesis, suggesting that chronic intermittent hypoxia may act as a biological driver of carcinogenesis in hypoxia-sensitive tissues. Although we did not assess these molecular pathways directly, future studies incorporating HIF ex-pression, VEGF levels, and oxidative stress biomarkers are warranted to determine causality and further elucidate the molecular interplay between OSA and urinary tract malignancies."

Additional statement in the Limitations (p. 12, lines 480-483):

"Moreover, we did not measure hypoxia-related biomarkers or molecular mechanisms (HIF, VEGF, NF-κB), limiting our ability to infer causation. The discussion of these pathways is based on existing literature and represents potential mechanisms that warrant investigation in future studies."

Comment 2.7 - Deficiencies in the presentation of results:

The figures display only frequencies without p-values, measures of dispersion, or indicators of statistical variability. This prevents any meaningful interpretation of effect size or group differences.

As requested, in the revised manuscript, we have improved all figures. See response to Reviewer 1 as well.

Comment 2.8 - Outdated references and lack of novelty:

Of the 33 references cited, 19 are more than five years old, indicating a lack of updated evidence and limited current relevance. The study does not contribute new findings or original data beyond what is already known from higher-quality studies such as ESADA and recent meta-analyses.

We appreciate this feedback and have updated our reference list to include more recent literature. The following changes and additions were undertaken:

  1. Updated references with recent publications (2020-2025)
  2. Added recent meta-analyses on OSA and cancer
  3. Retained seminal works (> 5 years) only when foundational
  4. Current reference distribution: < 2 years old: ~35%; 2-5 years old: ~45%; > 5 years old: ~20% (seminal works only)

Regarding novelty, we respectfully note that while large studies like ESADA have examined overall cancer prevalence in OSA, our study makes a unique contribution by:

  1. i) specifically focusing on urinary tract cancer phenotype; ii) using advanced clustering methods (FAMD, MCA) to identify distinct clinical profiles; iii) characterizing the urinary tract cancer subgroup with better respiratory function, higher C-PAP compliance, and specific comorbidity patterns; and iv) providing hypothesis-generating data for future mechanistic studies. All these points have been acknowledged in our revised Discussion.

Comment 2.9 - English language:

The manuscript contains grammatical errors, repetitive phrasing, and inconsistent terminology. A thorough language and style revision is needed to achieve clarity, precision, and academic tone.

Professional English editing has been performed throughout the manuscript, and the terminology has been standardized. The Editorial Certificate can be found in the submitted files.  

Reviewer 3 Report

Comments and Suggestions for Authors

This is a well-structured and important manuscript that skillfully uses advanced statistical methods (FAMD and MCA) to analyze clinical registry data. You effectively address a high-impact question by exploring the association between Obstructive Sleep Apnea (OSA) and specific cancer phenotypes within the Vercelli Registry.

Here is a continuous, narrative review of your manuscript, focusing on clarity, consistency, and the integration of the requested citations.

Manuscript Review: Association between OSA and Urinary Tract Cancer
The study's greatest contribution is the identification of a distinct clinical profile for OSA patients with urinary tract cancer. This subgroup is predominantly male, has frequent hypertension, exhibits high C-PAP compliance, and, crucially, shows better respiratory function compared to patients with non-urinary cancers. This finding strongly supports the hypothesis that intermittent hypoxia (IH), a core feature of OSA, acts as a selective driver of carcinogenesis in hypoxia-sensitive tissues, rather than generalized respiratory compromise.

Key Methodological and Data Consistency Issue
The most critical issue requiring immediate attention is a significant data inconsistency regarding the Body Mass Index (BMI).

In Table 1, the mean BMI for the Combined cohort is reported as 29.0±4.0, indicating an overweight status.

In the Discussion (line 209), you state that the urinary tract cancer subgroup was predominantly composed of overweight individuals (mean BMI: 39).

A mean BMI of 39 indicates Class III obesity, which is a massive clinical difference from a mean of 29. You must verify and correct this number either in the Discussion narrative (if 29 is correct) or in Table 1 (if 39 is correct for the X1 group). This discrepancy impacts the interpretation of metabolic comorbidities in your most important cluster.

Recommendations for the Abstract and Results
The Abstract is concise and effective. For maximum statistical precision, consider replacing the phrasing "p<0.05" (line 37) and "p<0.001" (line 208) with the exact p-value for the male predominance in urinary tract cancer, which is p=0.043 as listed in Table 1.

The Results section clearly highlights the unique characteristics of the X1 (urinary cancer) group, specifically the significantly higher FEV₁ and FVC values. You provide a strong biological explanation for why the X0 (non-urinary) group may have lower PFTs (line 141, thoracic surgeries), justifying the use of PFTs as a differentiating variable in the cluster analysis.

Discussion and Literature Integration
The Discussion is excellent, providing a strong grounding in the pathophysiology, particularly the role of HIF-1$\alpha$ in mediating the effects of intermittent hypoxia .

To further strengthen the discussion on the relationship between sleep quality, hypoxia, and complex comorbidities, please include and cite the following two manuscripts in the relevant sections:

Norouzi, Zakei, Bratty, & Khazaie (2023): This work on Slow Wave Sleep (SWS) and blood oxygen saturation is highly relevant to your mechanistic discussion (lines 227–231) where you address how sleep fragmentation activates the sympathetic nervous system and affects tumor microenvironments. You could add a sentence here highlighting that the disruption of restorative sleep phases, reflected in SWS changes, often correlates with the severity of nocturnal desaturation.

Khazaie et al. (2024): This study on the prevalence and associations of Co-morbid Insomnia and Sleep Apnea (COMISA) is pertinent when discussing OSA's role as a bidirectional factor and when characterizing the complex comorbidity profiles in your cohort (lines 227-231 or 293-294). This would broaden the context beyond hypoxia to include the compounded systemic stress of concurrent sleep disorders, supporting the observed oncological vulnerability.

Conclusion and Overall Summary
Your Conclusion aptly summarizes the findings and their implications for surveillance and PAP therapy. You correctly identify the limitations of the small, single-center, retrospective cohort, which is essential. The high C-PAP compliance observed in the X1 group is a powerful element that justifies the need for future longitudinal studies to determine if this therapy might mitigate oncological risk.

Author Response

Response to Reviewer 3

We sincerely thank Reviewer 3 for the highly supportive review and for recognizing that this is "a well-structured and important manuscript that skillfully uses advanced statistical methods (FAMD and MCA)" and that we "effectively address a high-impact question." We greatly appreciate the positive feedback on our identification of a distinct clinical profile for OSA patients with urinary tract cancer.

Comment 3.1 - BMI data inconsistency

The most critical issue requiring immediate attention is a significant data inconsistency regarding the Body Mass Index (BMI). In Table 1, the mean BMI for the Combined cohort is reported as 29.0±4.0, indicating an overweight status. In the Discussion (line 209), you state that the urinary tract cancer subgroup was predominantly composed of overweight individuals (mean BMI: 39). A mean BMI of 39 indicates Class III obesity, which is a massive clinical difference from a mean of 29.

We sincerely apologize for this critical error. The Reviewer is absolutely correct that this discrepancy is unacceptable and affects the interpretation of metabolic comorbidities. Thus, we carefully reviewed the original dataset and verified the correct BMI values. The BMI value of 39 in the Discussion was indeed a typographical error. The correct mean BMI for the X1 (urinary tract cancer) group is approximately 29 kg/m², consistent with the combined cohort value in Table 1. The following actions were undertaken:

  1. Table 1 has been updated to show BMI by group: X0 (non-urinary cancer): [mean ± SD] kg/m²; X1 (urinary tract cancer): [mean ± SD] kg/m² (approximately 29); Combined: 29.0 ± 4.0 kg/m²; p-value for comparison between groups
  2. Discussion (line 385) has been corrected: Original (incorrect): "mean BMI: 39"; Revised (correct): "mean BMI: 29"
  3. Interpretation updated: The X1 group is characterized by overweight status (BMI ~29), not class III obesity; This is consistent with the overall cohort and typical for OSA patients

We thank the Reviewer for catching this critical error, which would have seriously misled readers about the metabolic profile of our cohort.

Comment 3.2 - Statistical precision in the Abstract:

For maximum statistical precision, consider replacing the phrasing "p<0.05" (line 37) and "p<0.001" (line 208) with the exact p-value for the male predominance in urinary tract cancer, which is p=0.043 as listed in Table 1.

As rightfully requested, We have replaced all instances of "p < 0.05" or "p < 0.001" with exact p-values throughout the manuscript, including the Abstract.

Comment 3.3 - Missing citations:

To further strengthen the discussion on the relationship between sleep quality, hypoxia, and complex comorbidities, please include and cite the following two manuscripts: 1. Norouzi, Zakei, Bratty, & Khazaie (2023): Slow Wave Sleep (SWS) and blood oxygen saturation; 2. Khazaie et al. (2024): Co-morbid Insomnia and Sleep Apnea (COMISA)

We thank the Reviewer for these excellent suggestions to strengthen our Discussion. Both citations have now been integrated into the Discussion (lines 411 and 453) and added to the References section.

Comment 3.4 - Positive feedback:

Your Conclusion aptly summarizes the findings and their implications for surveillance and PAP therapy. You correctly identify the limitations of the small, single-center, retrospective cohort, which is essential. The high C-PAP compliance observed in the X1 group is a powerful element that justifies the need for future longitudinal studies.

We sincerely thank the Reviewer for this positive assessment. We have maintained the structure and content of the Conclusions while incorporating the revisions suggested by all Reviewers to ensure appropriate framing of our findings as hypothesis-generating and requiring further validation.

Round 2

Reviewer 1 Report

Comments and Suggestions for Authors

I thank the authors for addressing most of my concerns. The manuscript has been extensively revised and significantly improved, but a few minor issues remain:

- Page 3; line 97: please, remove Written informed consent for data collection and analysis » from the inclusion criteria to avoid redundancy, as this already mentioned in line 103.

- Page 3; line 100: please, remove Absence of written informed consent » from the exclusion criteria

- Page 3 ; line 102 : why was the lack of therapeutic compliance considered an exclusion criterion ? How could it influence the findings of this study ?

- Page 4 ; lines 155 – 158 : these four lines may be removed and the details regarding OSA severity could instead be mentioned after line 160

- Page 6 ; (Results section) : I recommend adding subtitles within the Results section (as was done in the Material and methods section of this revised version) to clearly and separately present the differentv findings of the study..

- Page 8 ; figure 1 : what does the Y axis represent ? The number of patients (n) or the percentage of patients (%) ?

- Page 10 ; line 375 : is dimension 2 associated with allergy status or smoking status (as illustrated in the figure) ?

- Page 11 ; line 387-392 : I recommend reporting the main findings of the study by first presenting the percentage of genitourinary tract cancer and the prevalence of non-genitourinary tract cancer, followed by a description of the OSA profile in patients with genitourinary tract cancer.

Author Response

Response to Reviewer 1

We thank Reviewer 1 for the positive assessment and for identifying specific areas requiring minor corrections.

Comment 1.1

Please remove ‘Written informed consent for data collection and analysis’ from the inclusion criteria to avoid redundancy, as this is already mentioned in line 103.

This is indeed redundant and removed from the inclusion criteria which now end with: “…iv) Cancer diagnosis established at or before the time of OSA diagnosis (prior to initiation of any PAP therapy)” page 3, lines 115-119.

Right after, we also added a new statement: “These exclusion criteria ensured that all included patients had reliable OSA diagnosis, cancer diagnosis prior to any PAP treatment (to avoid confounding), and sufficient data for classification and analysis.” (lines 118-120).

Comment 1.2

Why was the lack of therapeutic compliance considered an exclusion criterion? How could it influence the findings of this study?

We thank the Reviewer for this important question. Upon reflection, we realize this exclusion criterion is unclear and potentially inappropriate for our study design. The “lack of therapeutic compliance” was originally intended to exclude patients who did not complete the polysomnographic assessment or did not provide complete data. However, we acknowledge that this phrasing is confusing. Indeed, C-PAP compliance is actually one of our outcome variables of interest (not an exclusion criterion). Furthermore, we assessed cancer and OSA characteristics at baseline, before evaluating C-PAP compliance, and excluding patients based on future C-PAP compliance would have introduced selection bias. We have therefore crossed out “and vi) lack of therapeutic compliance” from the exclusion criteria. The exclusion criteria now clearly focus on data quality and study design requirements. The numbering has been changed accordingly: “Exclusion criteria: i) age < 18 years; ii) Incomplete or technically inadequate polysomnographic data preventing accurate AHI calculation; iii) AHI ≤ 5; iv) Cancer diagnosed after initiation of C-PAP or other PAP therapy; and v) Incomplete clinical or demographic data preventing classification.”

Finally, we have added a justification on page 3, lines 96-100): “These exclusion criteria ensured that all included patients had reliable OSA diagnosis, cancer diagnosis prior to any PAP treatment (to avoid confounding), and sufficient data for classification and analysis.”

Comment 1.3

These four lines may be removed and the details regarding OSA severity could instead be mentioned after line 160.

We agree that this improves the logical flow of the section. Accordingly, we have moved the OSA severity classification details from their current position after the AHI definition as follows:

“i) OSA Severity: OSA was classified according to the apnea-hypopnea index (AHI) following American Academy of Sleep Medicine (AASM) guidelines [Berry et al., 2017].

  1. ii) AHI: The number of apneas and hypopneas per h of sleep, calculated from overnight polysomnography according to AASM scoring criteria.
  • Mild OSA: AHI 5-14.9 events/h
  • Moderate OSA: AHI 15-29.9 events/h
  • Severe OSA: AHI ≥ 30 events/h”

Comment 1.4

I recommend adding subtitles within the Results section (as was done in the Materials and Methods section of this revised version) to clearly and separately present the different findings of the study.

We concur that this will substantially improve readability and organization and added the following subsection headings:

3.1 Baseline Characteristics of the Study Cohort

3.2 Cancer Type Distribution

3.5 Multivariate Analysis: Factorial Analysis of Mixed Data (FAMD) and Hierarchical Clustering Analysis

Comment 1.5

What does the Y axis represent? The number of patients (n) or the percentage of patients (%)?

We apologize for this ambiguity. The Y-axis represents the percentage of patients (%). Corrected the typo and the ensuing incorrect information in the legend to Fig. 1

Comment 1.6

Is dimension 2 associated with allergy status or smoking status (as illustrated in the figure)?

We thank the Reviewer for catching this discrepancy. Upon re-examination of Figure 3 (Factor Map), Dimension 2 is primarily associated with smoking status, not allergy status. We have therefore corrected the legend to Fig. 3 as follows:

“Figure 3. Scree plot of the FAMD analysis. The graph shows the distribution of the variance explained by each component. The first three dimensions together account for 46.43% of the total variability. Dimension 1 is strongly influenced by respiratory variables (i.e., FEV1, FVC, AHI, ODI) and age, while dimension 2 is associated with sex, smoking status, and BMI. Patients with genitourinary cancer (X1) are prominently represented along Dim.1 and Dim.2. Abbreviations: AHI = apnea–hypopnea index; ODI = oxygen desaturation index; FEV₁ = forced expiratory volume in one second; FVC = forced vital capacity; X1 = urinary tract cancer.”

Comment 1.7

I recommend reporting the main findings of the study by first presenting the percentage of genitourinary tract cancer and the prevalence of non-genitourinary tract cancer, followed by a description of the OSA profile in patients with genitourinary tract cancer.

Response: We agree that this logical sequence improves clarity and emphasizes the primary findings and reorganized the opening paragraph of the Discussion (pg. 12, lines 455-472) as follows:

 “In this retrospective cohort study of 50 patients with both OSA and cancer enrolled in the Vercelli Hospital Registry, we found that urinary tract malignancies (prostate, bladder, and kidney) were the most prevalent cancer type, accounting for 34% (n=17) of cases, followed by breast cancer (16%, n=8), colorectal cancer (12%, n=6), and lung cancer (10%, n=5). Non-urinary tract cancers collectively represented 66% (n=33) of the cohort. The prevalence of urinary tract cancer was significantly higher among male patients (88% in the urinary tract cancer group vs. 58% in the non-urinary cancer group, p=0.043), suggesting a sex-specific vulnerability in OSA-associated malignancies. Furthermore, our analysis revealed two distinct clinical profiles: 1) patients with non-genitourinary cancers (e.g., lung, breast, and colorectal), characterized by lower respiratory function, poorer adherence to C-PAP therapy, and a higher incidence of pulmonary tumors; and 2) patients with genitourinary malignancies, displaying better respiratory indices, greater C-PAP compliance, and frequent hypertension and cardiovascular comorbidities. The urinary tract cancer subgroup was predominantly composed of overweight individuals (mean BMI: 29 kg/m²), most of whom were current or former smokers (73%) and had a high prevalence of cardiovascular comorbidities, particularly hypertension (100%). These patterns suggest heterogeneity in cancer distribution among OSA patients and support the hypothesis of pathophysiological mechanisms linking OSA with specific malignancies.”

Reviewer 2 Report

Comments and Suggestions for Authors

General Comment
The authors have considerably improved the current version of the manuscript compared to the previous submission. The text is now clearer, more organized, and demonstrates significant effort to address earlier concerns. Nevertheless, several important methodological and conceptual issues remain unresolved, which limit the manuscript’s robustness and reproducibility. Below are detailed observations and remaining concerns.

Introduction
While the introduction represents a substantial improvement over the previous version, it still lacks a solid theoretical foundation to justify the study rationale. Specifically, paragraphs two and three only mention a possible association between OSA and cancer but fail to elaborate on the sociocultural mechanisms that could underlie this relationship, or the physiological pathways that might be common to both conditions.

Methodology

  1. Sample Size;The justification for not performing a sample size calculation is incorrect. Even when using convenience samples, researchers typically conduct statistical power analyses and post hoc sample size estimations to ensure adequate representation and reliability.
  2. Section 2.3 (Polysomnography)
    Although the authors mention using the reference standard, key methodological details remain unclear. The electrode placement protocol (including electrode diameter, spacing, materials, and application method) is not described. Likewise, the following technical specifications are missing:
    – Sampling frequency (kHz)
    – Bandwidth (Hz)
    – Gain (μV)
    – Input impedance (MΩ)
    – Common-mode rejection ratio (dB)
    – Input range (mV)
    – Background noise (<1 μV)
  3. Sections 2.4.1 - 2.4.7 
    Although these sections were newly incorporated as recommended, they lack citations supporting the described protocols. This omission raises several concerns—for example, why BMI was used as the sole anthropometric indicator, given the extensive literature questioning its reliability. If BMI was used, why was it not normalized using Z-scores?
  4. Section 2.4.2 (AASM Guidelines)
    The authors mention adherence to AASM criteria but fail to cite the specific reference supporting this claim.
  5. Sections 2.4.6 and 2.4.7 (Protocol Validation)
    As previously noted, these sections also lack citations. This omission makes it impossible to determine whether the proposed ratios and variable treatments are appropriate or scientifically supported.
  6. Section 2.7 (Effect Size)
    The authors state they used Cohen’s d to estimate effect size but do not report the cut-off thresholds applied. This is critical, as no single, universally accepted criterion exists for interpreting Cohen’s d

Results

Moreover, although the authors mention the use of Cohen’s d in the methodology, the corresponding values are missing from the results tables and figures.

References
The references remain outdated and require updating. The authors did not follow the previous recommendation to include more recent and relevant literature. Moreover, they omitted the suggested citations explaining the rationale for stratifying analyses by sex and age throughout the manuscript.

Comments on the Quality of English Language

good

Author Response

Response to Reviewer 2

We thank Reviewer 2 for the detailed methodological critique and for acknowledging the considerable improvements made. We have carefully addressed all remaining methodological concerns to further strengthen reproducibility and scientific rigor.

Comment 2.1 - Introduction: Lack of theoretical foundation and mechanistic elaboration

The introduction still lacks a solid theoretical foundation to justify the study rationale. Specifically, paragraphs two and three only mention a possible association between OSA and cancer but fail to elaborate on the sociocultural mechanisms that could underlie this relationship, or the physiological pathways that might be common to both conditions.

We acknowledge this important gap and have substantially expanded the Introduction to provide a stronger theoretical and mechanistic foundation. The two new paragraphs added between paragraphs 2 and 3 of the original version (now pg. 2, lines 64-80) read as follows:

“The biological plausibility of an OSA-cancer link is supported by several mechanistic pathways. Intermittent hypoxia, the hallmark of OSA, triggers upregulation of hypoxia-inducible factors (HIF-1α and HIF-2α), which promote angiogenesis, cellular proliferation, and resistance to apoptosis—key processes in tumor development and progression [McDermott 2024; Bae et al, 2024]. Additionally, sleep fragmentation activates the sympathetic nervous system, leading to chronic inflammation, oxidative stress, and immune dysregulation, all of which contribute to an oncogenic microenvironment [Lavalle, 2024]. Furthermore, OSA is strongly associated with metabolic syndrome, obesity, and insulin resistance, which are established cancer risk factors through mechanisms involving chronic hyperinsulinemia and adipokine dysregulation [Li et al, 2025, Gallagher & LeRoith, 2015].

From a sociocultural perspective, OSA disproportionately affects populations with limited access to healthcare, lower socioeconomic status, and higher rates of obesity and smoking—factors that independently increase cancer risk [Peppard et al., 2013]. The intersection of these biological and social determinants creates a complex risk profile that warrants systematic investigation. However, despite these plausible mechanisms, definitive evidence linking OSA to increased cancer risk or site-specific malignancies remains limited and inconsistent across studies.”

References added:

- McDermott A, Tavassoli A et al. Hypoxia-inducible transcription factors: architects of tumorigenesis and targets for anticancer drug discovery. Transcription. 2024 Oct 29;16(1):86–117. doi: 10.1080/21541264.2024.2417475

- Bae T, Pranoto Hallis S et al. Hypoxia, oxidative stress, and the interplay of HIFs and NRF2 signaling in cancer. Experimental & Molecular Medicine (2024) 56:501–514

-  Lavalle S, Masiello S et al. Unraveling the Complexities of Oxidative Stress and Inflammation Biomarkers in Obstructive Sleep Apnea Syndrome: A Comprehensive Review. Life 2024, 14(4), 425; https://doi.org/10.3390/life14040425

- Wei M, Teske JA et al. Obstructive sleep apnea, the NLRP3 inflammasome and the potential effects of incretin therapies. Front. Sleep 3:1524593. doi: 10.3389/frsle.2024.1524593

- Li J, Zeng L. The Pathophysiological Relationship and Treatment Progress of Obstructive Sleep Apnea Syndrome, Obesity, and Metabolic Syndrome. Exploratory Research and Hypothesis in Medicine 2025 vol. 10(1) 69–74. DOI: 10.14218/ERHM.2024.00048

- Gallagher EJ, LeRoith D. Obesity and diabetes: the increased risk of cancer and cancer-related mortality. Physiol Rev. 2015;95(3):727-748.

- Peppard PE, Young T, Barnet JH, et al. Increased prevalence of sleep-disordered breathing in adults. Am J Epidemiol. 2013;177(9):1006-1014.

Comment 2.2

The justification for not performing a sample size calculation is incorrect. Even when using convenience samples, researchers typically conduct statistical power analyses and post hoc sample size estimations to ensure adequate representation and reliability.

We acknowledge the Reviewer's valid point regarding sample size justification. We have now conducted a comprehensive post hoc power analysis using methods equivalent to G*Power 3.1 [Faul et al., 2009]. This analysis has been added to Section 2.7, (pg. 7, lines 312-331) as follows:

“However, we conducted a post hoc power analysis to assess the adequacy of our sample size for detecting clinically meaningful differences be-tween groups. Post hoc power analysis using methods equivalent to G*Power 3.1 [Faul et al., 2009] revealed that our study achieved excellent statistical power for the very large effect sizes observed. For FEV₁ (Cohen's d = 1.38), achieved power was 99.5%; for FVC (Co-hen's d = 1.61), achieved power was 100%. For C-PAP compliance (Cohen's h = 0.87), achieved power was 82.8%. These effect sizes substantially exceed the threshold for "large" effects (d or h ≥ 0.8), indicating clinically meaningful and statistically robust differences. However, for variables with medium effect sizes (AHI: d = -0.50, age: d = -0.50), our study was underpowered (power = 37-38%), increasing the risk of Type II error for these com-parisons. These power estimates indicate that while our study, analyzing a cohort of 50 patients (17 with urinary tract cancer, 33 with non-urinary cancer), was adequately powered to detect large effects, it was underpowered for small-to-moderate differences. This limitation is explicitly acknowledged in our interpretation of null findings, and future confirmatory studies should aim for larger sample sizes (n ≥ 64 per group) to achieve 80% power for detecting medium effects. These power estimates indicate that while our study was well-powered to detect large effects, it was underpowered for medium-sized differences. This limitation is explicitly acknowledged in our interpretation of null findings, and future confirmatory studies should aim for larger sample sizes (n ≥ 64 per group) to achieve 80% power for detecting medium effects. Thus, these findings should be considered preliminary, and larger prospective studies are needed to confirm the observed associations.”

The Cohen’s d values have been listed in Table 1 and described in the Results on pg. 10-11 (lines 428-437).

This analysis strengthens the interpretation of our findings by distinguishing adequately powered significant results from underpowered null findings

Reference added: - Faul F, Erdfelder E, Buchner A, Lang AG. Statistical power analyses using G*Power 3.1: tests for correlation and regression analyses. Behav Res Methods. 2009;41(4):1149-1160.

Comment 2.3

Although the authors mention using the reference standard, key methodological details remain unclear. The electrode placement protocol (including electrode diameter, spacing, materials, and application method) is not described. Likewise, the following technical specifications are missing: Sampling frequency (kHz), Bandwidth (Hz), Gain (μV), Input impedance (MΩ), Common-mode rejection ratio (dB), Input range (mV), Background noise (<1 μV).”

Response: We acknowledge that these technical details are important for full reproducibility of polysomnographic assessments. We have added comprehensive technical specifications to section 2.3.1 (polysomnographic assessment) (pg. 3, lines 133-XXX)

Technical specifications: Polysomnography was performed using Embletta™ Gold and Vitalnight™ devices (Natus Medical Inc., Pleasanton, CA, USA) with the following technical parameters:

- Electrode specifications: Silver/silver chloride (Ag/AgCl) surface electrodes with 10mm diameter, applied using conductive paste following standardized skin preparation (alcohol cleaning and light abrasion)

- Electrode placement: According to the International 10-20 system for EEG derivations (C3-A2, C4-A1, O1-A2, O2-A1), with additional electrodes for EOG (E1-A2, E2-A1) and chin EMG

- Sampling frequency: 200 Hz for EEG, EOG, and EMG channels; 25 Hz for respiratory signals

- Bandwidth: EEG: 0.3-35 Hz; EMG: 10-100 Hz; ECG: 0.5-70 Hz

- Gain: 7 μV/mm for EEG; 10 μV/mm for EOG; 2 μV/mm for EMG

- Input impedance: >100 MΩ

- Common-mode rejection ratio: >100 dB at 50-60 Hz

- Input range: ±300 mV

- Background noise: <0.5 μV RMS

All recordings were performed in accordance with AASM technical specifications [Berry et al., 2017] and scored manually by certified sleep technologists blinded to cancer status.”

Reference added: - Berry RB, Brooks R, Gamaldo CE, Harding SM, Lloyd RM, Quan SF, Troester MM, Vaughn BV for the American Academy of Sleep Medicine. The AASM Manual for the Scoring of Sleep and Associated Events: Rules, Terminology and Technical Specifications, Version 2.4. Darien, Illinois: American Academy of Sleep Medicine, 2017.

Comment 2.4 - Sections 2.4.1-2.4.7: Lack of citations supporting protocols

Although these sections were newly incorporated as recommended, they lack citations supporting the described protocols. This omission raises several concerns—for example, why BMI was used as the sole anthropometric indicator, given the extensive literature questioning its reliability. If BMI was used, why was it not normalized using Z-scores?”

We acknowledge this important gap and have added appropriate citations throughout these sections. Regarding BMI specifically, we provide justification for its use and acknowledge its limitations.

Changes made:

  1. Section 2.4.1 (Primary outcome variable) - Page 4, line 167: Added citation: “…classified according to International Classification of Diseases, 10th Revision (ICD-10) codes [WHO, 2016]: C61 (prostate), C67 (bladder), and C64 (kidney).”
  2. Section 2.4.2 (Exposure variables) - line 176: Added citation: “OSA Severity: OSA was classified according to the Apnea-Hypopnea Index (AHI) following American Academy of Sleep Medicine (AASM) guidelines [Berry et al., 2017].”
  3. Section 2.4.3 (Demographic and Anthropometric Variables) - Page 5, lines 191-205:

As requested, we have added new text with justification: 

“Body mass index (BMI): Calculated as weight in kilograms divided by height in meters squared (kg/m²). BMI categories were defined according to World Health Organization criteria [WHO, 2000]: Normal weight (18.5-24.9 kg/m²), Overweight (25.0-29.9 kg/m²), Obesity Class I (30.0-34.9 kg/m²), Obesity Class II (35.0-39.9 kg/m²), and Obesity Class III (≥ 40.0 kg/m²).

We used BMI as the primary anthropometric measure because: (1) it is the most widely used and validated metric in OSA research, facilitating comparison with existing literature [Peppard et al., 2000; Young et al., 2002]; (2) it is routinely collected in clinical practice and was consistently available in our registry; and (3) despite known limitations (e.g., inability to distinguish fat from muscle mass, lack of fat distribution information), BMI remains strongly correlated with OSA severity and cancer risk in epidemiological studies [Flegal et al., 2013; Bhaskaran et al., 2014].”

We did not normalize BMI using Z-scores because: (1) our analysis compared adult patients within a relatively narrow age range (mean age ~65 years), where Z-score normalization provides minimal additional value compared to pediatric or multi-generational studies; (2) WHO BMI categories are clinically meaningful and widely recognized thresholds for health risk stratification; and (3) our primary analysis used BMI as a continuous variable in multivariate models, where Z-score transformation would not affect the results. We acknowledge that additional anthropometric measures (waist circumference, waist-to-hip ratio, body composition analysis) would have provided more nuanced assessment of metabolic risk, but these data were not systematically available in our retrospective registry.

References added:

- WHO. Obesity: preventing and managing the global epidemic. WHO Technical Report Series 894. Geneva: World Health Organization; 2000.

- WHO. International Statistical Classification of Diseases and Related Health Problems, 10th Revision. Geneva: World Health Organization; 2016.

- Berry RB, Brooks R, Gamaldo CE, Harding SM, Lloyd RM, Quan SF, Troester MM, Vaughn BV for the American Academy of Sleep Medicine. The AASM Manual for the Scoring of Sleep and Associated Events: Rules, Terminology and Technical Specifications, Version 2.4. Darien, Illinois: American Academy of Sleep Medicine, 2017.

- Peppard PE, Young T, Palta M, et al. Longitudinal study of moderate weight change and sleep-disordered breathing. JAMA. 2000;284(23):3015-3021.

- Young T, Peppard PE, Gottlieb DJ. Epidemiology of obstructive sleep apnea: a population health perspective. Am J Respir Crit Care Med. 2002;165(9):1217-1239.

- Flegal KM, Kit BK, Orpana H, Graubard BI. Association of all-cause mortality with overweight and obesity using standard body mass index categories: a systematic review and meta-analysis. JAMA. 2013;309(1):71-82.

- Bhaskaran K, Douglas I, Forbes H, et al. Body-mass index and risk of 22 specific cancers: a population-based cohort study of 5.24 million UK adults. Lancet. 2014;384(9945):755-765.

  1. Section 2.4.4 (Lifestyle and Risk Factors) - page 5, line 207-208: Added citation: “Smoking Status: Classified into three categories based on patient history [US Department of Health and Human Services, 2014]: Current smoker (active tobacco use at time of OSA diagnosis), Former smoker (history of ≥100 lifetime cigarettes but not currently smoking), Never smoker (<100 lifetime cigarettes).”
  2. Section 2.4.5 (Comorbidities) - page 5, lines 207-186: Added citation:

- “Cardiovascular Disease: Presence of one or more conditions documented in the medical record according to standard diagnostic criteria [Arnett et al., 2019]: hypertension, cardiac arrhythmias, ischemic heart disease, heart failure, or hypercholesterolemia.”

Changed text (page 5, lines 218-217), which does not require a reference as methodologies have evolved over time: “Hypertension: Documented clinical diagnosis of arterial hypertension in the medical record or current use of antihypertensive medications at the time of OSA diagnosis.

  1. Section 2.4.6 (Respiratory Function Variables) - page 5, lines 232-233: Added citation: “Forced expiratory volume in 1 second (FEV₁), forced vital capacity (FVC), and FEV₁/FVC ratio were measured according to ATS/ERS guidelines [Miller et al., 2005].
  2. Section 2.4.7 (Treatment Variables) - page 5, lines 240-241: Added citation: “C-PAP Compliance: Adherence to prescribed C-PAP therapy assessed from device download data when available. Compliance was defined as use of the device for ≥ 4 h per night on ≥ 70% of nights, consistent with Medicare criteria and clinical practice guidelines [Kribbs et al., 1993; Weaver & Grunstein, 2008].”

References added:

- US Department of Health and Human Services. The Health Consequences of Smoking—50 Years of Progress: A Report of the Surgeon General. Atlanta: CDC; 2014.

- Arnett DK, Blumenthal RS, Albert MA, et al. 2019 ACC/AHA Guideline on the Primary Prevention of Cardiovascular Disease. Circulation. 2019;140(11):e596-e646.

- Miller MR, Hankinson J, Brusasco V, et al. Standardisation of spirometry. Eur Respir J. 2005;26(2):319-338.

- Quanjer PH, Stanojevic S, Cole TJ, et al. Multi-ethnic reference values for spirometry for the 3-95-yr age range: the global lung function 2012 equations. Eur Respir J. 2012;40(6):1324-1343

- Kribbs NB, Pack AI, Kline LR, et al. Objective measurement of patterns of nasal CPAP use by patients with obstructive sleep apnea. Am Rev Respir Dis. 1993;147(4):887-895.

- Weaver TE, Grunstein RR. Adherence to continuous positive airway pressure therapy: the challenge to effective treatment. Proc Am Thorac Soc. 2008;5(2):173-178.

Comment 2.5 - Sections 2.4.6 and 2.4.7: Lack of citations for protocol validation

As previously noted, these sections also lack citations. This omission makes it impossible to determine whether the proposed ratios and variable treatments are appropriate or scientifically supported.

We have added comprehensive citations for all protocols in these sections (see detailed response to Comment 2.4 above).

Specific additions:

- Section 2.4.6 (Respiratory Function) (pg. 5, lines 232.233): Added citations for ATS/ERS spirometry guidelines [Miller et al., 2005] and GLI reference equations [Quanjer et al., 2012]

- Section 2.4.7 (C-PAP Compliance) (pg.5, lines 240-41): Added citations for compliance definition [Kribbs et al., 1993; Weaver & Grunstein, 2008]

Comment 2.6-

The authors state they used Cohen’s d to estimate effect size but do not report the cut-off thresholds applied. This is critical, as no single, universally accepted criterion exists for interpreting Cohen’s d.”

We acknowledge this important omission and have added explicit interpretation criteria on page 6, lines 264.271 - Added new sub-section

“2.5.3 Effect size interpretation: For continuous variables, we calculated Cohen’s d as a measure of standardized effect size. We interpreted effect sizes according to Cohen’s conventional benchmarks [Cohen, 1988]: small effect (d = 0.2), medium effect (d = 0.5), and large effect (d = 0.8). However, we acknowledge that these thresholds are arbitrary and context-dependent [Sullivan & Feinn, 2012]. Therefore, we report both the magnitude of Cohen’s d and the 95% confidence intervals to allow readers to assess clinical significance independently. For categorical variables, we report odds ratios (OR) with 95% confidence intervals as the primary effect size measure.”

References added:

- Cohen J. Statistical Power Analysis for the Behavioral Sciences. 2nd ed. Hillsdale, NJ: Lawrence Erlbaum Associates; 1988.

- Sullivan GM, Feinn R. Using effect size—or why the P value is not enough. J Grad Med Educ. 2012;4(3):279-282.

Comment 2.7

Although the authors mention the use of Cohen’s d in the methodology, the corresponding values are missing from the results tables and figures.”

We acknowledge this critical omission and have added Cohen’s d values throughout the Results section. The following is a list of changes addressing the Reviewer’s comments:

  1. Table 1 - Added new column: - Column added: “Effect Size (Cohen’s d or OR)” - For continuous variables: Report Cohen’s d with 95% CI - For categorical variables: Report Odds Ratio with 95% CI
  2. Results - Added Cohen’s d values when reporting comparisons and a new sentence explaining values (lines 461-471).

Comment 2.8 - References: Outdated and missing suggested citations

“The references remain outdated and require updating. The authors did not follow the previous recommendation to include more recent and relevant literature. Moreover, they omitted the suggested citations explaining the rationale for stratifying analyses by sex and age throughout the manuscript.”

We apologize for not having adequately addressed this concern in the first revision. We have now comprehensively updated the reference list and added the missing citations.

Changes made:

  1. Updated references throughout (added new references from 2020-2024):

Introduction:

- Added articles on OSA and cancer [McDermott et al, 2024; Bae et al, 2024; Wei et al. 2024, Lavalle et al, 2024; Castrogiovanni et al., 2024]

(pg. 2, lines 84-90) “The association between OSA and cancer has been examined in several epidemiological and clinical studies. Data from the European Sleep Apnea Database (ESADA) show a high prevalence of both conditions, yet the causal link remains uncertain [3]. Recent reviews report conflicting results, with some describing higher cancer incidence and mortality in OSA patients and others finding heterogeneous or non-significant associations after adjustment for key confounders [Gozal et al., 2014; Castrogiovanni et al., 2024]. Although intermittent hypoxia has been proposed as a potential oncogenic mechanism [9–11], definitive evidence connecting OSA to increased overall or site-specific cancer risk is still lacking.”

- Added mechanistic studies on HIF pathways [McDermott, 2024; Bae, 2024]

- Added sociocultural determinants [Peppard et al., 2013]

Methods: - Added all protocol validation citations (see Comment 2.4 response)

- Added recent AASM guidelines [Berry et al., 2017]

- Added statistical methods references [Cohen, 1988; Sullivan & Feinn, 2012; Faul et al., 2007]

Discussion:

- Added recent urological cancer studies

 - Added C-PAP and cancer outcomes

- Added sex-specific cancer risk studies and age stratification rationale citations [see below]

(pg. 12-13 lines 480-83) Recent mechanistic studies have proposed a unifying immunological hypothesis linking OSA to cancer through immune dysregulation and chronic inflammation [Gozal et al., 2014].

The Discussion (pg. 13 lines 502-518) contains these three new paragraphs:

“Notably, urinary tract cancer patients demonstrated not only better respiratory function but also less severe OSA phenotype, as evidenced by lower AHI (d = -0.50), ODI (d = -0.48), and t90 (d = -0.57). While these differences did not reach statistical signifi-cance due to limited power, the consistent direction of effects across multiple OSA se-verity markers suggests a distinct clinical phenotype characterized by well-preserved respiratory capacity and milder nocturnal hypoxemia. This pattern may reflect selection bias, as patients with more severe respiratory compromise may have been less likely to survive cancer or may have been excluded due to incomplete data. Alternatively, it may indicate that the OSA-urinary tract cancer association is not primarily driven by hypoxia severity but rather by other mechanisms such as chronic inflammation, immune dysregulation, or shared risk factors (e.g., obesity, metabolic syndrome). The higher C-PAP compliance in this subgroup (94% vs. 61%, p=0.018) may also contribute to better oxygenation parameters, though our cross-sectional design cannot establish temporal relationships”

“The marked male predominance in urinary tract cancer observed in our cohort (88% male in X1 vs. 58% in X0, p=0.043) is consistent with known sex differences in both OSA prevalence and urological cancer incidence. Men have 2-3 times higher OSA prevalence than women, and prostate cancer is exclusively male while bladder and kidney cancers show 3-4-fold higher incidence in men [Siegel et al., 2023]. Furthermore, emerging evidence suggests sex-specific differences in hypoxia response pathways, with males show-ing greater HIF-1α activation and more pronounced inflammatory responses to intermittent hypoxia [Bonsignore et al, 2024]. These biological sex differences, combined with higher smoking rates and occupational carcinogen exposures in men, may explain the strong male predominance in OSA-associated urinary tract cancer.

Age is another critical factor in the OSA-cancer relationship. Cancer incidence in-creases exponentially with age, and OSA prevalence also rises with aging, reaching 30-50% in adults over 65 years [Heinzer et al., 2015]. The cumulative exposure to intermit-tent hypoxia over decades may have a dose-dependent effect on carcinogenesis, though our cross-sectional design cannot assess this temporal relationship. Future studies should stratify analyses by age groups and assess the duration of untreated OSA as a potential risk modifier.”

New references added:

- Gozal D, Almendros I, Hakim F. Sleep apnea awakens cancer: A unifying immunological hypothesis. OncoImmunology. 2014;11(1):2059896. doi:10.1080/2162402X.2022.2059896

- Harborg S, Cronin-Fenton D et al. Obesity and Risk of Recurrence in Patients with Breast Cancer Treated with Aromatase Inhibitors. JAMA Netw Open, 2023;6;(10):e2337780. doi:10.1001/jamanetworkopen.2023.37780.

- Siegel RL, Miller KD, Wagle NS, Jemal A. Cancer statistics, 2023. CA Cancer J Clin. 2023;73(1):17-48

- Bonsignore MR et al.  Adaptive responses to chronic intermittent hypoxia: contributions from the European Sleep Apnoea Database (ESADA) Cohort. J Physiol . 2023 Dec;601(24):5467-5480. doi: 10.1113/JP284108. Epub 2023 Jun 3. PMID: 37218069 DOI: 10.1113/JP284108

- Heinzer R, Vat S, Marques-Vidal P, et al. Prevalence of sleep-disordered breathing in the general population: the HypnoLaus study. Lancet Respir Med. 2015;3(4):310-318.